# Why the Metric Backbone
# Preserves Community Structure

**Maximilien Dreveton**
EPFL
maximilien.dreveton@epfl.ch

**Charbel Chucri**
EPFL
charbel.chucri@epfl.ch

**Matthias Grossglauser**
EPFL
matthias.grossglauser@epfl.ch

**Patrick Thiran**
EPFL
patrick.thiran@epfl.ch

## Abstract

The *metric backbone* of a weighted graph is the union of all-pairs shortest paths. It is obtained by removing all edges $(u, v)$ that are not on the shortest path between $u$ and $v$. In networks with well-separated communities, the metric backbone tends to preserve many inter-community edges, because these edges serve as bridges connecting two communities, but tends to delete many intra-community edges because the communities are dense. This suggests that the metric backbone would dilute or destroy the community structure of the network. However, this is not borne out by prior empirical work, which instead showed that the metric backbone of real networks preserves the community structure of the original network well. In this work, we analyze the metric backbone of a broad class of weighted random graphs with communities, and we formally prove the robustness of the community structure with respect to the deletion of all the edges that are not in the metric backbone. An empirical comparison of several graph sparsification techniques confirms our theoretical finding and shows that the metric backbone is an efficient sparsifier in the presence of communities.

## 1  Introduction

*Graph clustering* partitions the vertex set of a graph into non-overlapping groups, so that the vertices of each group share some typical pattern or property. For example, each group might be composed of vertices that interact closely with each other. Graph clustering is one of the main tasks in the statistical analysis of networks [4, 30].

In many scenarios, the observed pairwise interactions are weighted. In a *proximity graph*, these weights measure the degree of similarity between edge endpoints (e.g., frequency of interaction in a social network), whereas in a *distance graph*, they measure dissimilarity instead (for example, the length of segments in a road network or travel times in a flight network). To avoid confusion, we refer to the weights in a distance graph as *costs*, so that the cost of a path will be naturally defined as the sum of the edge costs on this path.[1]

Distance graphs obtained from real-world data typically violate the triangle inequality. More precisely, the shortest distance between two vertices in the graph is not always equal to the cost of the direct edge, but rather equal to the cost of an indirect path via other vertices. For example, the least expensive

---

[1]This additive property does not hold for similarity weights, but a proximity graph can be transformed into a distance graph by applying an isomorphic non-linear transformation on the weights [36].

38th Conference on Neural Information Processing Systems (NeurIPS 2024).

flight between two cities is often a flight including one or more layovers. An edge whose endpoints are connected by an indirect shorter path is called *semi-metric*; otherwise, it is called *metric*.

We obtain the *metric backbone* of a distance graph by removing all its semi-metric edges. It has been experimentally observed that the metric backbone retains only a small fraction of the original edges, typically between 10% and 30% in social networks [35]. Properties of the original graph that depend only on the shortest paths (such as connectivity, diameter, and betweenness centrality) are preserved by its metric backbone. Moreover, experiments on contact networks indicate that other properties (such as spreading processes and community structures) are also empirically well approximated by computing them on the metric backbone, rather than on the original graph [8].

The preservation of these properties by the backbone highlights a well-known empirical feature of complex networks: *redundancy*. This is the basis for *graph sparsification*: the task of building a sparser graph from a given graph so that important structural properties are approximately preserved while reducing storage and computational costs. Many existing network sparsifiers identify the statistically significant edges of a graph by comparing them to a null-model [34, 43]. These methods typically require hyperparameters and can leave some vertices isolated (by removing all edges from a given vertex). *Spectral sparsification* aims at preserving the spectral structure of a graph but also relies on a hyperparameter, and the edges in the sparsified graph might have different weights than in the original graph [37]. In contrast, the metric backbone is parameter-free and automatically preserves all properties linked to the shortest path structure. Moreover, all-pairs shortest paths can be efficiently computed [14, 33]. This makes the metric backbone an appealing mechanism for sparsification.

Among all the properties empirically shown to be well preserved by the metric backbone, the community structure is perhaps the most surprising. Indeed, if a network contains dense communities that are only loosely connected by a small number of inter-community edges, then all shortest paths between vertices in different communities must go through one of these edges. This suggests that these "bridge" edges are less likely to be semi-metric than intra-community edges, where the higher density provides shorter alternative paths.[2] This in turn implies that metric sparsification should thin out intra-community edges more than inter-community edges, thereby diluting the community structure. The central contribution of this paper is to show that this intuition is wrong.

To do so, we formally characterize the metric backbone of a weighted stochastic block model (wSBM). We assume that the vertices are separated into $k$ blocks (also referred to as clusters or communities). An edge between two vertices belonging to blocks $a$ and $b$ is present with probability $p_{ab}$ and is independent of the presence or absence of other edges. This edge, if present, has a weight sampled from a distribution with cdf $F_{ab}$, and this weight represents the cost of traveling through this edge. We denote by $p_{ab}^{\mathrm{mb}}$ the probability that an edge between a vertex in block $a$ and a vertex in block $b$ is present in the backbone.[3] Under loose assumptions on the costs distributions $F_{ab}$ and on the probabilities $p_{ab}$, we show that $p_{ab}^{\mathrm{mb}}/p_{cd}^{\mathrm{mb}} = (1 + o(1))p_{ab}/p_{cd}$ for every $a, b, c, d \in [k]$. This shows that the metric backbone thins the edge set approximately uniformly and, therefore, preserves the community structure of the original graph. Moreover, we also prove that a spectral algorithm recovers almost exactly the communities.

We also conduct numerical experiments with several graph clustering algorithms and datasets to back up these theoretical results. We show that all clustering algorithms achieve similar accuracy on the metric backbone as on the original network. These simulations, performed on different types of networks and with different clustering algorithms, generalize the experiments of [8], which are restricted to contact networks and the Louvain algorithm. Another type of graph we consider are graphs constructed from data points in a Euclidean space, typically by a kernel similarity measure between $q$-nearest neighbors ($q$-NN) [15]. This is an important technique for graph construction, with applications in non-linear embedding and clustering. Although $q$ controls the sparsity of this embedding graph, this procedure is not guaranteed to produce only metric edges, and varying $q$ often impacts the clustering performance. By investigating the metric backbone of $q$-NN graphs, we notice that it makes clustering results more robust against the value of $q$. Consequently, leveraging graph sparsification alongside $q$-NN facilitates graph construction.

---

[2] In fact, this intuitive observation is at the core of one of the very first community detection algorithms [17].

[3] The events $\mathbb{1}\{(u, v)$ is metric$\}$ and $\mathbb{1}\{(w, x)$ is metric$\}$ are in general not independent, but the probability that $(u, v)$ is metric depends only on the cluster assignment of vertices $u$ and $v$.

The paper is structured as follows. We introduce the main definitions and theoretical results on the metric backbone of wSBMs in Section 2. We discuss these results in Section 3 and compare them with the existing literature. Sections 4 and 5 are devoted to numerical experiments and applications. Finally, we conclude in Section 6.

**Code availability**    We provide the code used for the experiments: `https://github.com/Charbel-11/Why-the-Metric-Backbone-Preserves-Community-Structure`

**Notations**    The notation $1_n$ denotes the vector of size $n \times 1$ whose entries are all equal to one. For any vector $\pi \in \mathbb{R}^n$ and any matrix $B \in \mathbb{R}^{n \times m}$, we denote by $\pi_{\min} = \min_{a \in [n]} \pi_a$ and $B_{\min} = \min_{a,b} B_{ab}$, and similarly for $\pi_{\max}$ and $B_{\max}$. Given two matrices $A$ and $B$ of the same size, we denote by $A \odot B$ the entry-wise matrix product (*i.e.,* their Hadamard product). $A^T$ is the transpose of a matrix $A$. For a vector $\pi$, we denote $\mathrm{diag}(\pi)$ the diagonal matrix whose diagonal element $(a, a)$ is $\pi_a$.

The indicator of an event $A$ is denoted $\mathbb{1}\{A\}$. Binomial and exponential random variables are denoted by $\mathrm{Bin}(n, p)$ and $\mathrm{Exp}(\lambda)$, respectively. The uniform distribution over an interval $[a, b]$ is denoted $\mathrm{Unif}(a, b)$. Finally, given a cumulative distribution function $F$, we write $X \sim F$ for a random variable $X$ sampled from $F$, *i.e.,* $\mathbb{P}(X \leq x) = F(x)$, and we denote by $f$ the pdf of this distribution. We write whp (with high probability) for events with probability tending to 1 as $n \to \infty$.

## 2    The Metric Backbone of Weighted Stochastic Block Models

### 2.1    Definitions and Main Notations

Let $G = (V, E, c)$ be an undirected weighted graph, where $V = [n]$ is the set of vertices, $E \subseteq V \times V$ is the set of undirected edges, and $c \colon E \to \mathbb{R}_+$ is the cost function. The *cost* of a path $(u_1, \cdots, u_p)$ is naturally defined as $\sum_{q=1}^{p-1} c(u_q, u_{q+1})$, and a *shortest path* between $u$ and $v$ is a path of minimal cost starting from vertex $u$ and finishing at vertex $v$. We define the *metric backbone* as the union of all shortest paths of $G$.

**Definition 1.**    The *metric backbone* of a weighted graph $G = (V, E, c)$ where $c$ represents the edge cost is the subgraph $G^{\mathrm{mb}} = (V, E^{\mathrm{mb}}, c^{\mathrm{mb}})$ of $G$, where $E^{\mathrm{mb}} \subseteq E$ is such that $e \in E^{\mathrm{mb}}$ if and only if $e$ belongs to a shortest path from two vertices in $G$, and $c^{\mathrm{mb}} \colon E^{\mathrm{mb}} \to \mathbb{R}_+; e \mapsto c(e)$ is the restriction of $c$ to $E^{\mathrm{mb}}$.

We investigate the structure of the metric backbone of weighted random graphs with community structure. We generate these graphs as follows. Each vertex $u \in [n]$ is randomly assigned to the cluster $a \in [k]$ with probability $\pi_a$. We denote by $z_u \in [k]$ the cluster of vertex $u$. Conditioned on $z_u$ and $z_v$, an edge $(u, v)$ is present with probability $p_{z_u z_v}$, independently of the presence or absence of other edges. If an edge $(u, v)$ is present, it is assigned a cost $c(u, v)$. The cost $c(u, v)$ is sampled from $F_{z_u z_v}$ where $F = (F_{ab})_{1 \leq a, b \leq k}$ denotes a collection of cumulative distribution functions such that $F_{ab} = F_{ba}$. This defines the *weighted stochastic block model*, and we denote $(z, G) \sim \mathrm{wSBM}(n, \pi, p, F)$ with $G = ([n], E, c)$, $z \in [k]^n$ and

$$\mathbb{P}(z) = \prod_{u=1}^{n} \pi_{z_u},$$

$$\mathbb{P}(E \mid z) = \prod_{1 \leq u < v \leq n} p_{z_u z_v}^{\mathbb{1}\{\{u,v\} \in E\}} (1 - p_{z_u z_v})^{\mathbb{1}\{\{u,v\} \notin E\}},$$

$$\mathbb{P}(c \mid E, z) = \prod_{\{u,v\} \in E} \mathbb{P}(c(u, v) \mid z_u, z_v),$$

and $c(u, v) \mid z_u = a, z_v = b$ is sampled from $F_{ab}$.

The wSBM is a direct extension of the standard (non-weighted) SBM into the weighted setting. SBM is the predominant benchmark for studying community detection and establishing theoretical guarantees of recovery by clustering algorithms [1]. Despite its shortcomings, such as a low clustering coefficient, the SBM is analytically tractable and is a useful model for inference purposes [32].

Throughout this paper, we will make the following assumptions.

**Assumption 1** (Asymptotic scaling). The edge probabilities $p_{ab}$ between blocks $a$ and $b$ can depend on $n$, such that $p_{ab} = B_{ab}\rho_n$ with $\rho_n = \omega(n^{-1}\log n)$ and $B_{ab}$ is independent of $n$. Furthermore, the number of communities $k$, the matrix $B$, the probabilities $\pi_a$ and the cdf $F_{ab}$ are all fixed (independent of $n$). We also assume $\pi_{\min} > 0$ and $B_{\min} > 0$.

To rule out some issues, such as edges with a zero cost,[4] we also assume that the probability distributions of the costs have no mass at 0. More precisely, we require that the cumulative distribution functions $F_{ab}$ verify $F_{ab}(0) = 0$ and $F'_{ab}(0) = \lambda_{ab} > 0$, where $F'$ denotes the derivative of $F$ (*i.e.,* the pdf). The first condition ensures that the distribution has support $\mathbb{R}_+$ and no mass at 0, and the second ensures that, around a neighborhood of 0, $F_{ab}$ behaves as the exponential distribution $\mathrm{Exp}(\lambda_{ab})$ (or as the uniform distribution $\mathrm{Unif}([0, \lambda_{ab}^{-1}])$).

**Assumption 2** (Condition on the cost distributions). The costs are sampled from continuous distributions, and there exists $\Lambda = (\lambda_{ab})_{a,b}$ with $\lambda_{ab} > 0$ such that $F_{ab}(0) = 0$ and $F'_{ab}(0) = \lambda_{ab}$.

We define the following matrix

$$T = [\Lambda \odot B]\,\mathrm{diag}(\pi), \tag{2.1}$$

where $B$ and $\Lambda$ are defined in Assumptions 1 and 2. Finally, we denote by $\tau_{\min}$ and $\tau_{\max}$ the minimum and maximum entries of the vector $\tau = T1_k$.

**Remark 1.** Assume that $\Lambda = \lambda 1_k 1_k^T$ with $\lambda > 0$. We notice that $\tau_a = \lambda \sum_b \pi_b B_{ab}$. Denote by $\bar{d} = (\bar{d}_1, \cdots, \bar{d}_k)$ the vector whose $a$-entry $\bar{d}_a$ is the expected degree of a vertex in community $a$. We have $\bar{d}_a = n \sum_b \pi_b p_{ab}$. Then $\tau_{\min} = \lambda \bar{d}_{\min}(n\rho_n)^{-1}$ and $\tau_{\max} = \lambda \bar{d}_{\max}(n\rho_n)^{-1}$, where $\bar{d}_{\min}$ and $\bar{d}_{\max}$ are the minimum and maximum entries of $\bar{d}$.

## 2.2 Cost of Shortest Paths in wSBMs

Given a path $(u_1, \cdots, u_p)$, recall that its cost is $\sum_{q=1}^{p-1} c(u_q, u_{q+1})$, and the *hop count* is the number of edges composing the path (that is, the hop count of $(u_1, \cdots, u_p)$ is $p - 1$).

For two vertices $u, v \in V$, we denote by $C(u, v)$ the cost of the shortest path from $u$ to $v$.[5] The following proposition provides asymptotics for the cost of shortest paths in wSBMs.

**Proposition 1.** *Let $(z, G) \sim \mathrm{wSBM}(n, \pi, p, F)$. Suppose that Assumptions 1 and 2 hold and let $\tau_{\min}$ and $\tau_{\max}$ be defined following Equation (2.1). Then, for two vertices $u$ and $v$ chosen uniformly at random in blocks $a$ and $b$, respectively. We have whp*

$$(\tau_{\max})^{-1} \leq \frac{n\rho_n}{\log n} C(u, v) \leq (\tau_{\min})^{-1}.$$

To prove Proposition 1, in the first stage, we simplify the problem by assuming exponentially distributed weights. We then analyze two first passage percolation (FPP) processes, originating from vertices $u$ and $v$, respectively. Using the memoryless property of the exponential distribution, we analyze two first passage percolation (FPP) processes, originating from vertex $u$ and $v$, respectively. Each FPP explores the nearest neighbors of its starting vertex until it reaches $q$ neighbors. As long as $q = o(\sqrt{n})$, the two FPP processes remain disjoint (with high probability). Thus, the cost $C(u, v)$ is lower-bounded by the sum of (i) the cost of the shortest path from $u$ to its $q$-nearest neighbor and of (ii) the cost of the shortest path from $v$ to its $q$-nearest neighbor. On the contrary, when $q = \omega(\sqrt{n})$, the two FPPs intersect, revealing a path from $u$ to $v$, and the cost of this path upper-bounds the cost $C(u, v)$ of the shortest path from $u$ to $v$. In the second stage, we extend the result to general weight distributions by noticing that the edges belonging to the shortest paths have very small costs. Moreover, Assumption 2 yields that the weight distributions behave as an exponential distribution in a neighborhood of 0. We can thus adapt the coupling argument of [23] to show that the edge weights distributions do not need to be exponential, as long as Assumption 2 is verified. The proof of Proposition 1 is provided in Section A.

---

[4]An edge $(u, v) \in E$ such that $c(u, v) = 0$ yields the possibility of teleportation from vertex $u$ to vertex $v$.
[5]We notice that, given a wSBM, whp there is only one shortest path.

## 2.3 Metric Backbone of wSBMs

Let $(z, G) \sim \mathrm{wSBM}(n, \pi, p, F)$ and denote by $G^{\mathrm{mb}}$ the metric backbone of $G$. Choose two vertices $u$ and $v$ uniformly at random, and notice that the probability that the edge $(u, v)$ is present in $G^{\mathrm{mb}}$ depends only on $z_u$ and $z_v$, and not on $z_w$ for $w \notin \{u, v\}$. Denote by $p_{ab}^{\mathrm{mb}}$ the probability that an edge between a vertex in community $a$ and a vertex in community $b$ appears in the metric backbone, *i.e.,*

$$p_{ab}^{\mathrm{mb}} = \mathbb{P}\left((u, v) \in G^{\mathrm{mb}} \mid z_u = a, z_v = b\right).$$

The following theorem shows that the ratio $\frac{p_{ab}^{\mathrm{mb}}}{p_{ab}}$ scales as $\Theta\left(\frac{\log n}{n \rho_n}\right)$.

**Theorem 1.** *Let $(z, G) \sim \mathrm{wSBM}(n, \pi, p, F)$ and suppose that Assumptions 1 and 2 hold. Let $\tau_{\min}$ and $\tau_{\max}$ be defined after Equation* (2.1). *Then*

$$(1 + o(1)) \frac{\lambda_{ab}}{\tau_{\max}} \leq \frac{n \rho_n}{\log n} \frac{p_{ab}^{\mathrm{mb}}}{p_{ab}} \leq (1 + o(1)) \frac{\lambda_{ab}}{\tau_{\min}}.$$

We prove Theorem 1 in Appendix B.1. Theorem 1 shows that the metric backbone maintains the same proportion of intra- and inter-community edges as in the original graph. We illustrate the theorem with two important examples.

**Example 1.** Consider a weighted version of the planted partition model, where for all $a, b \in [k]$ we have $\pi_a = 1/k$ and

$$B_{ab} = \begin{cases} p_0 & \text{if } a = b, \\ q_0 & \text{otherwise,} \end{cases}$$

where $p_0$ and $q_0$ are constant. Assume that $\Lambda = \lambda 1_k 1_k^T$ with $\lambda > 0$. Using Remark 1, we have $\tau_{\min} = \tau_{\max} = \lambda k^{-1} (p_0 + (k - 1)q_0)$, and Theorem 1 states that

$$p^{\mathrm{mb}} = (1 + o(1)) \frac{k p_0}{p_0 + (k - 1)q_0} \frac{\log n}{n} \quad \text{and} \quad q^{\mathrm{mb}} = (1 + o(1)) \frac{k q_0}{p_0 + (k - 1)q_0} \frac{\log n}{n}.$$

In particular, $\frac{p^{\mathrm{mb}}}{q^{\mathrm{mb}}} = (1 + o(1)) \frac{p_0}{q_0}$.

**Example 2.** Consider a stochastic block model with edge probabilities $p_{ab} = B_{ab} \rho_n$ such that the vertices of different communities have the same expected degree $\bar{d}$. If $\Lambda = \lambda 1_k 1_k^T$, then

$$p_{ab}^{\mathrm{mb}} = (1 + o(1)) \frac{B_{ab}}{\bar{d}} \frac{\log n}{n}.$$

## 2.4 Recovering Communities from the Metric Backbone

In this section, we prove that a spectral algorithm on the (weighted) adjacency matrix of the metric backbone of a wSBM asymptotically recovers the clusters whp. Given an estimate $\hat{z} \in [k]^n$ of the clusters $z \in [k]^n$, we define the loss as

$$\mathrm{loss}(z, \hat{z}) = \frac{1}{n} \inf_{\sigma \in \mathrm{Sym}(k)} \mathrm{Ham}\left(z, \sigma \circ \hat{z}\right),$$

where $\mathrm{Ham}$ denotes the Hamming distance and $\mathrm{Sym}(k)$ is the set of all permutations of $[k]$.

---

**Algorithm 1:** Spectral Clustering on the weighted adjacency matrix of the metric backbone

**Input:** Graph $G$, number of clusters $k$
**Output:** Predicted community memberships $\hat{z} \in [k]^n$

1. Denote $W^{\mathrm{mb}} \in \mathbb{R}_+^{n \times n}$ the weighted adjacency matrix of the metric backbone $G^{\mathrm{mb}}$ of $G$
2. Let $W^{\mathrm{mb}} = \sum_{i=1}^n \sigma_i u_i u_i^T$ be an eigen-decomposition of $W^{\mathrm{mb}}$, with eigenvalues ordered in decreasing absolute value ($|\sigma_1| \geq \cdots \geq |\sigma_n|$) and eigenvectors $u_1, \cdots, u_n \in \mathbb{R}^n$
3. Denote $U = [u_1, \cdots, u_k] \in \mathbb{R}^{n \times k}$ and $\Sigma = \mathrm{diag}(\sigma_1, \cdots, \sigma_k)$
4. Let $\hat{z} \in [k]^n$ be a $(1 + \epsilon)$-approximate solution of $k$-means performed on the rows of $U \in \mathbb{R}^{n \times k}$

---

The following theorem states that, as long as the matrix $T$ defined in (2.1) is invertible, the loss of spectral clustering applied on the metric backbone asymptotically vanishes whp.

**Theorem 2.** *Let* $(z, G) \sim \mathrm{wSBM}(n, \pi, p, F)$ *and suppose that Assumptions 1 and 2 hold. Let* $\mu$ *be the minimal absolute eigenvalue of the matrix* $T$ *defined in (2.1). Moreover, assume that* $\tau_{\max} = \tau_{\min}$ *and* $\mu \neq 0$. *Then, the output* $\hat{z}$ *of Algorithm 1 on* $G$ *verifies whp*

$$\mathrm{loss}(z, \hat{z}) = O\left(\frac{1}{\mu^2 \log n}\right).$$

We prove Theorem 2 in Appendix B.2. We saw in Example 1 and 2 that the condition $\tau_{\max} = \tau_{\min}$ is verified in several important settings. The additional assumption $\mu \neq 0$ (equivalent to $T$ being invertible) also often holds: in the planted partition model of Example 1, $T$ is invertible if $p_0 \neq q_0$.

## 3 Comparison with Previous Work

The metric backbone has been introduced under different names, such as the *essential subgraph* [29], the *transport overlay network* [41], or simply the *union of shortest path trees* [39]. In this section, we discuss our contribution with respect to closely related earlier works.

### 3.1 Computing the Metric Backbone

Computing the metric backbone requires solving the All Pairs Shortest Path (APSP) problem, a classic and fundamental problem in computer science. Simply running Dijkstra's algorithm on each vertex of the graph solves the APSP in $O(nm + n^2 \log n)$ worst-case time, where $m = |E|$ is the number of edges in the original graph [14], whereas [29] proposed an algorithm running in $O(nm' + n^2 \log n)$ worst-case time, where $m'$ is the number of edges in the metric backbone. APSP has also been studied in weighted random graphs in which the weights are independent and identically distributed [20, 16]. In particular, the APSP can be solved in $O(n^2)$ time with high probability on complete weighted graphs whose weights are drawn independently and uniformly at random from $[0, 1]$ [33].

However, practical implementations of APSP can achieve faster results. For example, in [24], an empirical observation regarding the low hop count of shortest paths is leveraged to compute the metric backbone efficiently. Although exact time complexity is not provided, the implementation scales well for massive graphs, such as a Facebook graph with 190 million nodes and 49.9 billion edges, and the empirical running time appears to be linear with the number of edges [24, Table 1 and Figure 5]. Additionally, our simulations reveal that some popular clustering algorithms such as spectral and subspace clustering have higher running times than computing the metric backbone.

### 3.2 First-Passage Percolation in Random Graphs

To study the metric backbone theoretically, we need to understand the structure of the shortest path between vertices in a random graph. This classical and fundamental topic of probability theory is known as first-passage percolation (FPP) [19]. The paper [23] originally studied the weights and hop counts of shortest paths on the complete graph with iid weights. This was later generalized to Erdős-Rényi graphs and configuration models (see, for example, [28, 12] and references therein).

Closer to our setting, [25] studied the FPP on inhomogeneous random graphs. Indeed, SBMs are a particular case of inhomogeneous random graphs, for which the set of vertex types is discrete (we refer to [7] for general statements on inhomogeneous random graphs). Assuming that the edge weights are independent and $\mathrm{Exp}(\lambda)$-distributed, [25] established a central limit theorem of the weight and hop count of the shortest path between two vertices chosen uniformly at random among all vertices. Using the notation of Section 2, this result implies that $\frac{n\rho_n}{\log n}C(u, v)$ converges in probability to $\tilde{\tau}^{-1}$, where $\tilde{\tau}$ is the Perron-Frobenius eigenvalue of $\lambda B \mathrm{diag}(\pi)$.

The novelty of our work is two-fold. First, we allow different cost distributions for each pair of communities, whereas all previous works in FPP on random graphs assume that the costs are sampled from a single distribution. Furthermore, we examine the cost of a path between two vertices, $u$ and $v$, chosen *uniformly at random among vertices in block* $a$ *and in block* $b$, respectively. This differs from previous work (and, in particular, [25]) in which vertices $u$ and $v$ are *selected uniformly at random among all vertices*. As a result, even for a single cost distribution, Proposition 1 cannot be obtained directly from [25, Theorem 1.2]. This difference is key, as this proposition is required to establish Theorem 1.

The closest result to Theorem 1 appearing in the literature is [39, Corollary 1]; it establishes a formula for the probability $p_{uv}^{\mathrm{mb}}$ that an edge between two vertices $u$ and $v$ exists in the metric backbone of a random graph whose edge costs are iid. This previous work does not focus on community structure, so the costs are sampled from a single distribution. More importantly, the expression of $p_{uv}^{\mathrm{mb}}$ given by [39, Theorem 2 and Corollary 1] is mainly of theoretical interest (and we use it in the proof of Theorem 1). Indeed, understanding the asymptotic behavior of $p_{uv}^{\mathrm{mb}}$ requires a complete analysis of the cost $C(u, v)$ of the shortest path between $u$ and $v$. [39] propose such an analysis only in one simple scenario (namely, a complete graph with iid exponentially distributed costs).

## 4 Experimental Results

In this section, we test whether the metric backbone preserves a graph community structure in various real networks for which a ground truth community structure is known (see Table 2 in Appendix C.1 for an overview).

As in many weighted networks, such as social networks, the edge weights represent a measure (e.g., frequency) of interaction between two entities over time, we need to preprocess these proximity graphs into distance graphs. More precisely, given the original (weighted or unweighted) graph $G = (V, E, s)$, where the weights measure the similarities between pairs of vertices, we define the proximity $p(u, v)$ of vertices $u$ and $v$ as the *weighted Jaccard similarity* between the neighborhoods of $u$ and $v$, *i.e.,*

$$p(u, v) \;=\; \frac{\sum_{w \in \mathrm{Nei}(u) \cap \mathrm{Nei}(v)} \min\{s(u, w), s(v, w)\}}{\sum_{u \in \mathrm{Nei}(u) \cup \mathrm{Nei}(v)} \max\{s(u, w), s(v, w)\}},$$

where $\mathrm{Nei}(u) = \{w \in V : (u, w) \in E\}$ denotes the neighborhood of $u$, *i.e.,* the vertices connected to $u$ by an edge. If $G$ is unweighted ($s(e) = 1$ for all $e \in E$), we simply recover the Jaccard index $\frac{|\mathrm{Nei}(u) \cap \mathrm{Nei}(v)|}{|\mathrm{Nei}(u) \cup \mathrm{Nei}(v)|}$. We note that other choices for normalization could have been made, such as the Adamic-Adar index [2]. We refer to [10] for an in-depth discussion of similarity and distance indices.

Once the proximity graph $G = (V, E, p)$ has been computed, we construct the distance graph $D = (V, E, c)$ where $c \colon E \to \mathbb{R}_+$ is such that

$$\forall e \in E : \; c(e) \;=\; \frac{1}{p(e)} - 1.$$

This is the simplest and most commonly used method for converting a similarity to a distance [36].

We then compute the set $E^{\mathrm{mb}}$ of metric edges of the distance graph $D$, and let $G^{\mathrm{mb}} = (V, E^{\mathrm{mb}}, p^{\mathrm{mb}})$ where $p^{\mathrm{mb}} \colon E^{\mathrm{mb}} \to [0, 1]; e \mapsto p(e)$ is the restriction of $p$ to $E^{\mathrm{mb}}$.

We will also consider the two following sparsifications (with the corresponding restrictions of $c$ to the sparsified edge sets) to compare the resulting community structure:

- the *threshold graph* $G^\theta = (V, E^\theta, p^\theta)$, where an edge $e \in E$ is kept in $E^\theta$ iff $p(e) \geq \theta$;
- the graph $G^{\mathrm{ss}} = (V, E^{\mathrm{ss}}, p^{\mathrm{ss}})$ obtained by *spectral sparsification* on $G$. We use the Spielman-Srivastava sparsification [37], implemented in the PyGSP package [13].

For both threshold and spectral sparsification, we tune the hyperparameters so that the number of edges kept is the same as in the metric backbone: $|E^{\mathrm{mb}}| = |E^\theta| = |E^{\mathrm{ss}}|$. We provide in Table 3 (in Appendix) some statistics on the percentage of edges remaining in the sparsified graphs.

For each proximity graph $G$ and its three sparsifications $G^{\mathrm{mb}}$, $G^\theta$, and $G^{\mathrm{ss}}$, we run a graph clustering algorithm to obtain the respective predicted clusters $\hat{z}$, $\hat{z}^{\mathrm{mb}}$, $\hat{z}^\theta$ and $\hat{z}^{\mathrm{ss}}$. We show, in Figure 1, the *adjusted Rand index*[6] (ARI) obtained between the ground truth communities and the predicted clusters for three widely used graph clustering algorithms: *Bayesian* algorithm [31], *Leiden* algorithm [38] and *spectral clustering* [40]. We use the *graph-tool* implementation for the *Bayesian* algorithm, with an exponential prior for the weight distributions. The *Leiden* algorithm is implemented at

---

[6]The Rand index (RI) measures similarity between two clusterings based on pair counting. This index is 'adjusted for chance' because a random clustering could lead to a large RI. The ARI is a modification of the RI such that two random clusterings (resp., two identical clusterings) give an ARI of 0 (resp., of 1) [22].

 For *spectral clustering*, we assume that the algorithm knows the correct number of clusters in advance, and we use the implementation from *scikit-learn*.[7]

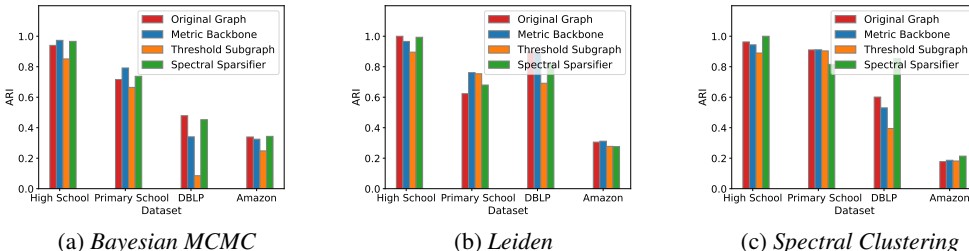

| (a) *Bayesian MCMC* | (b) *Leiden* | (c) *Spectral Clustering* |

Figure 1: Effect of sparsification on the performance of clustering algorithms on various data sets. We observe that the metric backbone and the spectral sparsification retain equally well the community structure across all data sets and for all clustering algorithms tested. Thresholding often yields several disconnected components of small sizes, impacting the performance of clustering algorithms on $G^\theta$.

We highlight the difference between the metric backbone and the threshold subgraph of the *Primary school* data set in Figure 2. We observe, in Figure 2a, that the edges in red (which are present in the backbone but not in the threshold graph) are mostly inter-community edges. On the contrary, in Figure 2b, the blue edges (which are present in the threshold graph but not in the backbone) are mostly intra-community edges. Despite this difference, the metric backbone retains the information about the community structure, as shown in Figure 1.

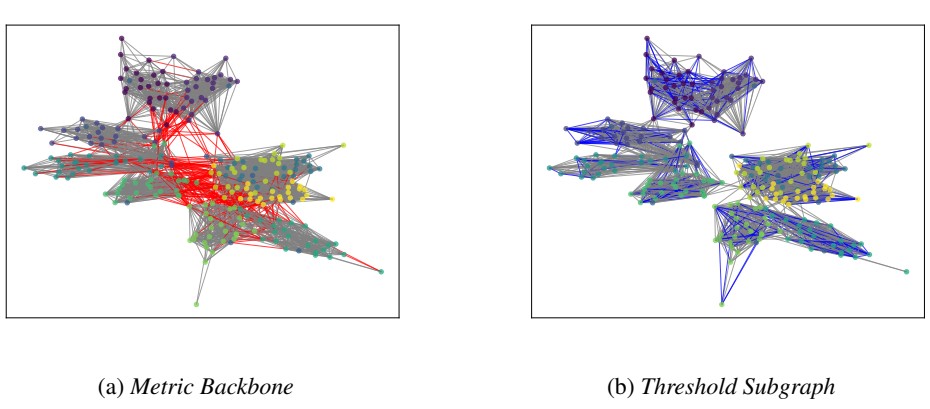

| (a) *Metric Backbone* | (b) *Threshold Subgraph* |

Figure 2: Graphs obtained from *Primary school* data set, after taking the metric backbone (Figure 2a) and after thresholding (Figure 2b), are drawn using the same layout. Vertex colors represent the true clusters. Edges present in the metric backbone but not in the threshold graph are highlighted in red. Edges present in the threshold graph, but not in the metric backbone, are highlighted in blue.

## 5 Application to Graph Construction Using $q$-NN

In a large number of machine-learning tasks, data does not originally come from a graph structure, but instead from a cloud of points $x_1, \cdots, x_n$ where each $x_u$ belongs to a metric space (say $\mathbb{R}^d$ for simplicity). The goal of *graph construction* is to discover a proximity graph $G = ([n], E, p)$ from the original data points. Graph construction is commonly done through the $q$-nearest neighbors ($q$-NN). Given a similarity function $\mathrm{sim}\colon \mathbb{R}^d \times \mathbb{R}^d \to \mathbb{R}_+$ that quantifies the resemblance between two data points, we define the set $\mathcal{N}(u, q)$ of $q$-nearest neighbors of $u \in [n]$. More precisely, $\mathcal{N}(u, q)$ is the subset of $[n] \backslash \{u\}$ with cardinality $q$ such that for all $v \in \mathcal{N}(u, q)$ and for all $w \notin \mathcal{N}(u, q)$ we have

$$\mathrm{sim}(x_u, x_v) \geq \mathrm{sim}(x_u, x_w).$$

---

[7]We also note that the threshold graph $G^\theta$ is often not connected. Hence, we ignored all components comprising less than 5 nodes before running spectral clustering.

The edge set $E$ is composed of all pairs of vertices $(u, v)$ such that $u \in \mathcal{N}(v, q)$ or $v \in \mathcal{N}(u, q)$,[8] and the proximity $p_{uv}$ associated with the edge $(u, v)$ is $(s_{uv} + s_{vu})/2$, where

$$s_{uv} = \begin{cases} \text{sim}(x_u, x_v) & \text{if } v \in \mathcal{N}(u, q), \\ 0 & \text{otherwise.} \end{cases}$$

In the following, we use the Gaussian kernel similarity $\text{sim}(x_u, x_v) = \exp\left(-\frac{\|x_u - x_v\|^2}{d_K^2(x_u)}\right)$, where $d_K(x_i)$ is the Euclidean distance between $x_u$ and its $q$-NN. In Appendix C.2, we provide results using another similarity measure.

We investigate the effect of graph sparsification on graphs built by $q$-NN. We sample $n = 10000$ images from MNIST [26] and FashionMNIST [42], and use the full HAR [3] dataset ($n = 10299$). From the sampled data points, we build the $q$-NN graph $G_q$, as well as its *metric backbone* $G_q^{\text{mb}}$ and its *spectral sparsification* $G_q^{\text{ss}}$. We then compare the performance of two clustering algorithms, *spectral clustering* and *Poisson learning*. *Poisson learning* is a semi-supervised graph clustering algorithm and was recently shown to outperform other graph-based semi-supervised algorithms [9]. Results of spectral clustering using another similarity measure are presented in the Appendix.

We compare the ARI of clustering algorithms on $q$-NN graphs and its sparsifications (metric backbone and spectral sparsification) for various choices of the number of nearest neighbors $q$. The results are shown in Figure 3 (for spectral clustering) and Figure 4 (for Poisson-learning). Unlike the spectral sparsifier, the metric backbone retains a high ARI across all choices of $q$. Interestingly, the performance on the original graph often decreases with $q$, which is a hyperparameter of the graph construction step. Applying a clustering algorithm on the metric backbone comes with the two advantages of significantly reducing the number of edges in the graph and of making its performance robust against the choice of the hyperparameter $q$. Indeed, a larger value of $q$ creates more edges but with a higher distance (cost), which are therefore more likely to be non-metric.

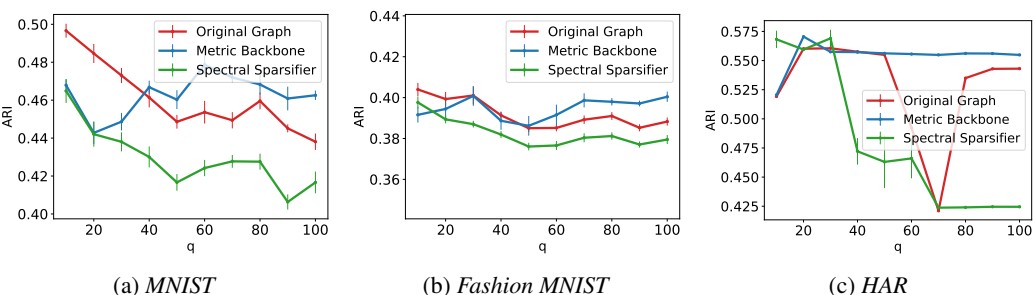

(a) *MNIST*  (b) *Fashion MNIST*  (c) *HAR*

Figure 3: Performance of *spectral clustering* on subsets of MNIST, FashionMNIST datasets, and on the HAR dataset. The ARI is averaged over 10 trials; error bars show the standard error of the mean.

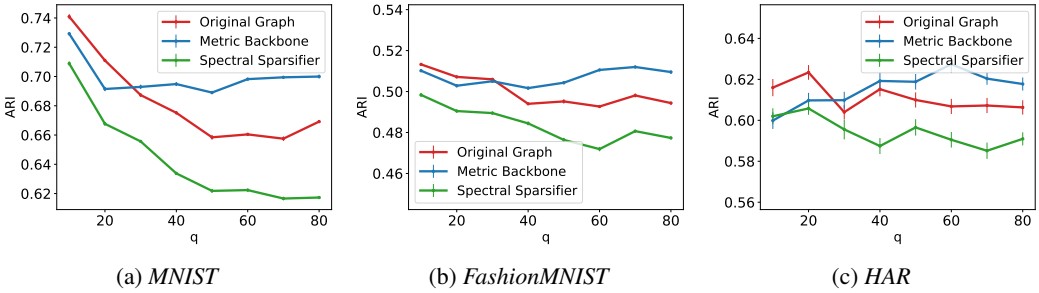

(a) *MNIST*  (b) *FashionMNIST*  (c) *HAR*

Figure 4: Performance of *Poisson learning* on subsets of MNIST, FashionMNIST datasets, and the HAR dataset. The ARI is averaged over 100 trials, and error bars show the standard error of the mean.

Finally, we compare in Table 1 the ARI obtained on the $q$-nearest neighbor graph using $q = 10$ (as it is a common default choice) with the metric backbone graph of a $q$-nearest neighbor graph with

---

[8]In other words, an edge $(u, v)$ is present if at least one of its two endpoints is in the $q$-nearest neighborhood of the other.

$q = \sqrt{n}/2$. Moreover, we compute an approximation of the metric backbone by sampling uniformly at random $2 \log n$ vertices and taking the union of the $2 \log n$ shortest-path trees rooted at each one of them instead of the union of all $n$ shortest-path trees. This produces a graph $\tilde{G}^{\mathrm{mb}}_{\sqrt{n}/2}$ whose edge set is a subset of the edges of the true metric backbone $G^{\mathrm{mb}}_{\sqrt{n}/2}$. We observe that $\tilde{G}^{\mathrm{mb}}_{\sqrt{n}/2}$ retains the community structure albeit being typically twice sparser than the 10-nearest neighbor graph $G_{10}$.

| Algorithm | Data set | | $G_{10}$ | $\tilde{G}^{\mathrm{mb}}_{\sqrt{n}/2}$ |
|---|---|---|---|---|
| Spectral clustering | MNIST | ARI | 0.533 | **0.563** $\pm$ 0.011 |
| | | Edges | $550,653$ | $373,379 \pm 3,018$ |
| | FashionMNIST | ARI | 0.411 | **0.425** $\pm$ 0.006 |
| | | Edges | $578,547$ | $272,063 \pm 857$ |
| | HAR | ARI | **0.519** | 0.492 $\pm$ 0.001 |
| | | Edges | $77,526$ | $37,535 \pm 977$ |
| Poisson learning | MNIST | ARI | 0.814 $\pm$ 0.015 | **0.821** $\pm$ 0.021 |
| | | Edges | $550,653$ | $373,379 \pm 3,018$ |
| | FashionMNIST | ARI | 0.526 $\pm$ 0.010 | **0.544** $\pm$ 0.014 |
| | | Edges | $578,547$ | $272,063 \pm 857$ |
| | HAR | ARI | 0.618 $\pm$ 0.039 | **0.636** $\pm$ 0.037 |
| | | Edges | $77,526$ | $37,535 \pm 977$ |

Table 1: Comparison of clustering on the *full* data sets. $G_{10}$ denotes the 10-nearest neighbors graph, and $\tilde{G}^{\mathrm{mb}}_{\sqrt{n}/2}$ denotes the approximate metric backbone of the $\sqrt{n}/2$-nearest neighbor graph. We approximate the metric backbone by sampling only $2 \log n$ shortest-path trees.

**Additional discussion** The performance of clustering algorithms on the $q$-nearest neighbor graph $G_q$ tends to decrease when $q$ increases. Indeed, a larger $q$ introduces many low-similarity edges, which can act as noise. Spectral sparsification preserves the spectral properties of the graph, but this becomes ineffective if the spectral properties of $G_q$ are insufficient to recover the communities (as attested by the poor performance of spectral clustering for large values of $q$ in Figures 3a and 8a). However, when the performance of spectral clustering on $G_q$ remains stable as $q$ is varied, so does the performance of spectral clustering on the spectral sparsified graph $G_q^{\mathrm{ss}}$ (as seen in Figures 3b and 8b). In contrast, the metric backbone preserves the shortest paths rather than spectral properties. Because the shortest paths are robust to the addition of numerous low-similarity edges,[9] the performance of clustering algorithms on the metric backbone $G_q^{\mathrm{mb}}$ remains stable when $q$ increases. This holds regardless of whether the performance on the original graph $G_q$ is stable or decreases with increasing $q$. Finally, sparsified graphs obtained by metric sparsification are more consistent across different values of $q$ than those obtained via spectral sparsification. For instance, on the MNIST dataset with Gaussian kernel similarity, the metric backbone $G_{30}^{\mathrm{mb}}$ and $G_{40}^{\mathrm{mb}}$ have both approximately 70k edges, with 64k edges in common. In contrast, the spectrally sparsified graphs $G_{30}^{\mathrm{ss}}$ and $G_{40}^{\mathrm{ss}}$, also with around 70k edges each, share only 22k edges in common.

## 6  Conclusion

The metric backbone plays a crucial role in preserving several essential properties of a network. Notably, the metric backbone effectively preserves the network community structure, although many inter-community edges belong to shortest paths. In this paper, we have specifically proven that the metric backbone preserves the community structure in weighted stochastic block models. Moreover, our numerical experiments emphasize the performance of the metric backbone as a powerful graph sparsification tool. Furthermore, the metric backbone can serve as a preprocessing step for graph construction employing $q$-nearest neighbors, alleviating the sensitivity associated with selecting the hyperparameter $q$ and producing sparser graphs.

---

[9]Consider, for example, an adversary adding an arbitrary number of edges with a cost of $\omega\left(\frac{\log n}{n \rho_n}\right)$ in a wSBM. Proposition 1 proves that this addition does not affect the metric backbone, as none of these high-cost edges will be included in any shortest path.

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

# A  Proof of Proposition 1

Let us set up some notations before proving Proposition 1. We let $(z, G) \sim \mathrm{wSBM}(n, \pi, p, f)$ where the model parameters verify Assumptions 1 and 2, and we denote by $c$ and $\tilde{c}$ the cost and the extended cost associated with $G$, respectively. We let $\Gamma_1, \cdots, \Gamma_k$ be the $k$ communities, *i.e.,* $\Gamma_a = \{w \in [n]\colon z_w = a\}$, and we denote by $n_1, \cdots, n_k$ their respective sizes, *i.e.,* $n_a = |\Gamma_a|$. As $k = \Theta(1)$, the concentration of multinomial distributions ensures that $n_a = (1 + o(1))\pi_a n$ whp.

## A.1  Particular case of exponentially distributed costs

*Proof of Proposition 1 for exponentially distributed costs.* We firstly assume that $F_{ab} \sim \mathrm{Exp}(\lambda_{ab})$ with $\lambda_{ab} > 0$. It is immediate to notice that $(f_{ab})_{ab}$ verify Assumption 2.

Let $u, v$ be two vertices chosen uniformly at random in $\Gamma_a$ and $\Gamma_b$ respectively, so that $z_u = a$ and $z_v = b$. To explore the neighborhood of vertex $u$, we consider a first passage percolation (FPP) on $G$ starting from $u$. More precisely, for any $t > 0$, we denote by $\mathcal{B}(u, t) = \{w \in V\colon C(u, w) \leq t\}$ the set of vertices within a cost $t$ of $u$, and by

$$C_u(q) = \min\{t \geq 0\colon |\mathcal{B}(u, t)| \geq q + 1\},$$

the cost going from $u$ to its $q$-th nearest neighbor (with the convention that $\min \emptyset = \infty$). In particular, $\mathcal{B}(u, C_u(q))$ is the set of the $q$ nearest neighbors of $u$. Let $U(q) = (U_1(q), \cdots, U_k(q))$ be the collection of sets such that $U_\ell(q)$ is the set of vertices that are in the $q$ nearest neighborhood of $u$ and in community $\ell$, *i.e.,*

$$U_\ell(q) = \mathcal{B}(u, C_u(q)) \cap \Gamma_\ell.$$

Finally, we denote by $S(u, q)$ the matrix whose entry $S_{\ell\ell'}(u, q)$ is equal to the number of edges going from $U_\ell(q)$ to $\Gamma_{\ell'} \backslash U_{\ell'}(q)$.

The outline of the proof is as follows. In a first step (i), we study the FPP starting from vertex $u$ to obtain upper and lower bounds on the cost $C_u(\sqrt{n \log n})$ for going from $u$ to its $\sqrt{n \log n}$-nearest neighbor[10]. More precisely, we will establish that

$$\frac{n\rho_n}{\log n} C_u\left(\sqrt{n \log n}\right) \leq \frac{1 + o(1)}{2\tau_{\min}} \tag{A.1}$$

holds whp. By symmetry, the same upper bound also holds for $C_v(\sqrt{n \log n})$.

Next, we will show in step (ii) that $\mathcal{B}(u, \sqrt{n \log n}) \cap \mathcal{B}(v, \sqrt{n \log n}) \neq \emptyset$ whp. Together with the upper bound (A.1), we can then upper bound the cost $C(u, v)$ of the shortest path from $u$ to $v$. Indeed, let $w_1 \in \mathcal{B}(u, \sqrt{n \log n}) \cap \mathcal{B}(v, \sqrt{n \log n})$ (such a $w_1$ exists whp by step (ii)), and consider the path $\mathcal{P} = u \to w_1 \to v$, where $u \to w_1$ and $w_1 \to v$ denote the shortest path from $u$ to $w_1$ and from $w_1$ to $v$, respectively. Because the cost $C(u, v)$ of the shortest path from $u$ to $v$ is upper-bounded by the cost of the path $\mathcal{P}$, we have whp

$$C(u, v) \leq C(u, w_1) + C(w_1, v) \leq \frac{1 + o(1)}{\tau_{\min}} \frac{\log n}{n\rho_n},$$

where the second inequality holds because $w_1 \in \mathcal{B}(u, \sqrt{n \log n}) \cap \mathcal{B}(v, \sqrt{n \log n})$, which enables to use the upper bound (A.1). This establishes the desired upper bound on $C(u, v)$.

To lower-bound $C(u, v)$, we will establish that

$$\frac{1 + o(1)}{2\tau_{\max}} \leq \frac{n\rho_n}{\log n} C_u\left(\sqrt{\frac{n}{\log n}}\right) \tag{A.2}$$

holds whp. Again, a similar bound holds for $C_v\left(\sqrt{n/\log n}\right)$ by symmetry.

Finally, we will show in step (iii) that $\mathcal{B}\left(u, C_u(\sqrt{n/\log n})\right) \cap \mathcal{B}\left(v, C_v(\sqrt{n/\log n})\right) = \emptyset$ whp. Combining this with the lower bound (A.2), we conclude then that whp

$$C(u, v) \geq \frac{1 + o(1)}{\tau_{\max}} \frac{\log n}{n\rho_n}.$$

---

[10]Because $\sqrt{n \log n}$ is not necessarily integer we should write $\lceil \sqrt{n \log n} \rceil$ but we drop the $\lceil$ and $\rceil$ to avoid overburdening the notations.

**(i) Upper and lower bounds on $C_u(\sqrt{n/\log n})$ and $C_u(\sqrt{n\log n})$.** In this paragraph, we will establish (A.1) and (A.2).

A key property of the FPP is to notice that, conditioned on $S(u, \cdot) = (S(u, 1), \cdots, S(u, n))$, the random numbers $C_u(q+1) - C_u(q)$ are independent and exponentially distributed, such that

$$C_u(q+1) - C_u(q) \mid S(u, \cdot) \sim \mathrm{Exp}\left(\sum_{\ell, \ell'} \lambda_{\ell\ell'} S_{\ell\ell'}(u, q)\right). \tag{A.3}$$

Indeed, $C_u(q+1) - C_u(q)$ is the difference in the cost of traveling from $u$ to its $(q+1)$-th nearest-neighbor minus the cost from $u$ and its $q$-th nearest neighbor. In other words,

$$C_u(q+1) - C_u(q) \mid S(u, \cdot) = \min_{\substack{w \in U_c(q) \\ w' \in \Gamma_{c'} \backslash U_{c'}(q)}} c_{ww'}, \tag{A.4}$$

with $c_{ww'} \sim \mathrm{Exp}(\lambda_{cc'})$. Statement (A.3) follows because $\min\{\mathrm{Exp}(\lambda), \mathrm{Exp}(\mu)\} \sim \mathrm{Exp}(\lambda + \mu)$.

Let $q \le \sqrt{n \log n}$. The number of edges $S_{\ell\ell'}(u, q)$ between the sets $U_\ell(q)$ and $\Gamma_{\ell'} \backslash U_{\ell'}(q)$ is binomially distributed such that

$$S_{\ell\ell'}(u, q) \sim \mathrm{Bin}\left(|U_\ell(q)| \times |\Gamma_{\ell'} \backslash U_{\ell'}(q)|, p_{cc'}\right).$$

The number of vertices in $\Gamma_{\ell'}$ is $n_{\ell'} = \Theta(n)$ and the number of vertices in $|U_{\ell'}(q)|$ verifies $|U_{\ell'}(q)| \le q \ll n$. Therefore,

$$\mathbb{E}[S_{\ell\ell'}(u, q)] = |U_\ell(q)| (n_{\ell'} - |U_{\ell'}(q)|) p_{\ell\ell'} = (1 + o(1)) |U_\ell(q)| n_{\ell'} p_{\ell\ell'}.$$

Hence $\mathbb{E}[S_{\ell\ell'}(u, q)] \gg 1$, and the concentration of binomial distributions ensure that whp

$$S_{\ell\ell'}(u, q) = (1 + o(1)) |U_\ell(q)| (n_{\ell'} - |U_{\ell'}(q)|) p_{\ell\ell'} = (1 + o(1)) |U_\ell(q)| n_{\ell'} p_{\ell\ell'}. \tag{A.5}$$

Therefore, using $n_{\ell'} = (1 + o(1))\pi_{\ell'} n$ and $p_{\ell\ell'} = B_{\ell\ell'} \rho_n$, we have from (A.4)

$$\begin{aligned} \mathbb{E}[C_u(q+1) - C_u(q) \mid S(u, \cdot)] &= \frac{1 + o(1)}{\sum_\ell |U_\ell(q)| \sum_{\ell'} n_{\ell'} \lambda_{\ell\ell'} p_{\ell\ell'}} \\ &= \frac{1 + o(1)}{n\rho_n \sum_\ell |U_\ell(q)| \sum_{\ell'} \pi_{\ell'} \lambda_{\ell\ell'} B_{\ell\ell'}}. \end{aligned}$$

We notice that $\sum_l |U_\ell(q)| \sum_{\ell'} \pi_{\ell'} \lambda_{\ell\ell'} B_{\ell\ell'} = U^T(q) T 1_k$ where $T = (\Lambda \odot B) \mathrm{diag}(\pi)$ is the operator defined in (2.1). Then, using $\sum_\ell |U_\ell(q)| = q$, we have

$$\tau_{\min} q \le \sum_\ell |U_\ell(q)| \sum_{\ell'} \pi_{\ell'} \lambda_{\ell\ell'} B_{\ell\ell'} \le \tau_{\max} q. \tag{A.6}$$

Therefore,

$$\frac{1 + o(1)}{n\rho_n \tau_{\max} q} \le \mathbb{E}[C_u(q+1) - C_u(q) \mid S(u, \cdot)] \le \frac{1 + o(1)}{n\rho_n \tau_{\min} q}. \tag{A.7}$$

This bound is uniform in $S(u, \cdot)$. Thus, using the total law of probability and summing over all $1 \le q \le \sqrt{n \log n}$ leads to

$$\frac{1 + o(1)}{2\tau_{\max}} \le \frac{n\rho_n}{\log n} \mathbb{E} C_u(\sqrt{n \log n}) \le \frac{1 + o(1)}{2\tau_{\min}}, \tag{A.8}$$

where we used $\sum_{q=1}^{\sqrt{n \log n}} q^{-1} = 2^{-1}(\log n + \log \log n) + \Theta(1)$.

Let us now upper-bound the variance of $C_u(\sqrt{n \log n})$. By the law of total variance, we have

$$\mathrm{Var}\left[C_u(\sqrt{n \log n})\right] = \mathbb{E}\left[\mathrm{Var}\left[C_u(\sqrt{n \log n}) \mid S(u, \cdot)\right]\right] + \mathrm{Var}\left[\mathbb{E}\left[C_u(\sqrt{n \log n}) \mid S(u, \cdot)\right]\right]. \tag{A.9}$$

The first term on the right-hand side of (A.9) can be upper bounded by proceeding similarly as for the expectation. Indeed, we have

$$\mathrm{Var}[C_u(q+1) - C_u(q) \mid S(u, \cdot)] \le \frac{1 + o(1)}{n^2 \rho_n^2 \tau_{\min}^2 q^2},$$

and the independence of $C_u(q+1) - C_u(q)$ conditioned on $S(u, \cdot)$ leads to

$$\mathrm{Var}\left[C_u(\sqrt{n \log n}) \mid S(u, \cdot)\right] = \sum_{q=1}^{\sqrt{n \log n} - 1} \mathrm{Var}\left(C_u(q+1) - C_u(q) \mid S(u, \cdot)\right)$$

$$\leq \frac{1 + o(1)}{n^2 \rho_n^2 \tau_{\min}^2} \sum_{q=1}^{\sqrt{n \log n}} q^{-2}$$

$$\leq \frac{1 + o(1)}{n^2 \rho_n^2 \tau_{\min}^2} \frac{\pi^2}{6}. \tag{A.10}$$

To upper bound the second term on the right-hand side of (A.9), we notice that

$$\mathrm{Var}\left[\mathbb{E}\left[C_u(\sqrt{n \log n}) \mid S(u, \cdot)\right]\right] = \mathrm{Var}\left[\sum_{q=1}^{\sqrt{n \log n} - 1} \mathbb{E}\left[C_u(q+1) - C_u(q) \mid S(u, \cdot)\right]\right]$$

$$= \sum_{q=1}^{\sqrt{n \log n} - 1} \mathrm{Var}\left[\mathbb{E}\left[C_u(q+1) - C_u(q) \mid S(u, \cdot)\right]\right],$$

where the first line holds by the linearity of the conditional expectation, and the second one by the independence of $C_u(q+1) - C_u(q)$ conditioned on $S(u, \cdot)$. Moreover, recall that $\mathrm{Var}(X) \leq (b-a)^2/4$ if $a \leq X \leq b$. Hence, using the upper and lower bound on $C_u(q+1) - C_u(q)$ obtained in (A.7) leads to

$$\mathrm{Var}\left[\mathbb{E}\left[C_u(q+1) - C_u(q) \mid S(u, \cdot)\right]\right] \leq \frac{1 + o(1)}{4}\left(\frac{1}{n \rho_n \tau_{\min} q} - \frac{1}{n \rho_n \tau_{\max} q}\right)^2$$

$$\leq \frac{1 + o(1)}{4}\left(\frac{1}{n \rho_n \tau_{\min} q}\right)^2,$$

and therefore

$$\mathrm{Var}\left[\mathbb{E}\left[C_u(\sqrt{n \log n}) \mid S(u, \cdot)\right]\right] \leq \frac{1 + o(1)}{4 n^2 \rho_n^2 \tau_{\min}^2} \frac{\pi^2}{6}. \tag{A.11}$$

Combining (A.10) and (A.11) into (A.9) provides

$$\mathrm{Var}\left[C_u(\sqrt{n \log n})\right] \leq \frac{5}{4} \frac{(1 + o(1))\pi^2}{6 \tau_{\min}^2 n^2 \rho_n^2}. \tag{A.12}$$

This upper bound (A.12) ensures that $\mathrm{Var}\left[C_u(\sqrt{n \log n})\right] = o\left(\left(\mathbb{E}\left[C_u(\sqrt{n \log n})\right]\right)^2\right)$, and therefore an application of Chebyshev's inequality ensures that

$$\frac{1 + o(1)}{2 \tau_{\max}} \leq \frac{n \rho_n}{\log n} C_u\left(\sqrt{n \log n}\right) \leq \frac{1 + o(1)}{2 \tau_{\min}}$$

with high probability. Likewise, by symmetry for a FPP starting from $v$ instead of $u$, we find

$$\frac{1 + o(1)}{2 \tau_{\max}} \leq \frac{n \rho_n}{\log n} C_v\left(\sqrt{n \log n}\right) \leq \frac{1 + o(1)}{2 \tau_{\min}}.$$

This establishes (A.1). Moreover, we establish (A.2) by doing a slight modification of this proof. More precisely, we sum over all $q \leq \sqrt{n / \log n}$ instead of all $q \leq \sqrt{n \log n}$ in step (A.8). We obtain the same bound as in (A.8) because $\sum_{q=1}^{\sqrt{n / \log n}} q^{-1} = 2^{-1}(\log n - \log \log n) + \Theta(1)$.

**(ii)** $\mathcal{B}(u, C_u(\sqrt{n \log n})) \cap \mathcal{B}(v, C_v(\sqrt{n \log n})) \neq \emptyset$ **whp.** For ease of notations, let us shorten by $\mathcal{B}_u = \mathcal{B}(u, C_u(\sqrt{n \log n}))$ the set of the $\sqrt{n \log n}$ nearest neighbours of $u$ and by $\mathcal{B}_v(q)$ the set of the $q$-nearest neighbours of $v$. We also denote by $w_q$ the $q$-nearest neighbor of $v$. A key property is that the two FPPs (starting from vertex $u$ and starting from vertex $v$) are independent of each other as

long as they do not intersect. To make this rigorous, we denote by $Q$ the random variable counting the number of steps made by the FPP starting from $v$ without intersecting with $\mathcal{B}_u$, *i.e.,*

$$Q \;=\; \min\left\{q \in [n] \colon \mathcal{B}_u \cap \mathcal{B}_v(q) \neq \emptyset\right\}.$$

We now show that $\mathbb{P}\left(Q > q\right) \;=\; o(1)$ whenever $q \gg \sqrt{n/\log n}$, which implies that $\mathcal{B}_u \cap \mathcal{B}_v(\sqrt{n \log n}) \neq \emptyset$. Using [6, Lemma B.1], we have

$$\mathbb{P}\left(Q > m\right) \;=\; \mathbb{E}\left[\prod_{q=1}^{m} \mathbb{Q}^{(q)}\left(Q > q \,|\, Q > q-1\right)\right], \tag{A.13}$$

where $\mathbb{Q}^{(q)}$ denotes the conditional distribution given $\mathcal{B}_u$ and $\mathcal{B}_v(q)$. We further notice that

$$\mathbb{Q}^{(q)}\left(Q > q \,|\, Q > q-1\right) \;=\; 1 - \mathbb{Q}^{(q)}\left(Q = q \,|\, Q > q-1\right),$$

and that the event $\{Q > q - 1\}$ is equivalent to the event $\{w_1 \notin \mathcal{B}_u, \cdots, w_{q-1} \notin \mathcal{B}_u\}$, *i.e.,* the FPP starting from $v$ has not yet collided with the one starting from $u$. Conditionally on this event, all vertices within the same block have an equal probability of being chosen at the $q$-th step of the FPP. Therefore,

$$
\begin{aligned}
\mathbb{Q}^{(q)}\left(Q = q \,|\, Q > q-1\right) &\;=\; \sum_{\ell \in [k]} \mathbb{P}\left(w_q \in \mathcal{B}_u \,|\, w_q \in \Gamma_\ell, Q > q-1\right) \mathbb{P}\left(w_q \in \Gamma_\ell\right) \\
&\;=\; \sum_{\ell \in [k]} \frac{|\mathcal{B}_u \cap \Gamma_\ell|}{|\Gamma_\ell|} \mathbb{P}\left(w_q \in \Gamma_\ell\right).
\end{aligned}
$$

Because $\sum_{\ell \in [k]} |\mathcal{B}_u \cap \Gamma_\ell| = \sqrt{n \log n}$, we have that[11] $\max_\ell |\mathcal{B}_u \cap \Gamma_\ell| \geq k^{-1}\sqrt{n \log n}$. Moreover, $|\Gamma_\ell| \leq (1 + o(1))\pi_{\max} n$ whp and $\sum_{\ell \in [k]} \mathbb{P}\left(w_q \in \Gamma_\ell\right) = 1$. Therefore,

$$
\begin{aligned}
\mathbb{Q}^{(q)}\left(Q = q \,|\, Q > q-1\right) &\;\geq\; \frac{\max_{\ell \in [k]} |\mathcal{B}_u \cap \Gamma_\ell|}{\min_{\ell \in [k]} |\Gamma_\ell|} \sum_{\ell \in [k]} \mathbb{P}\left(w_q \in \Gamma_\ell\right) \\
&\;\geq\; \frac{1 + o(1)}{\pi_{\max}} \sqrt{\frac{\log n}{n}}.
\end{aligned}
$$

Going back to (A.13), this implies that

$$\mathbb{P}\left(Q > m\right) \;\leq\; \left(1 - \frac{1 + o(1)}{\pi_{\max}} \sqrt{\frac{\log n}{n}}\right)^{m},$$

which indeed goes to 0 when $m \gg \sqrt{n/\log n}$.

**(iii)** $\mathcal{B}(u, C_u(\sqrt{n/\log n})) \cap \mathcal{B}(v, C_v(\sqrt{n/\log n})) = \emptyset$ **whp.** We proceed similarly to step (ii) by considering the FPP starting from $v$, and denote by $w_q$ the $q$-nearest neighbor of $v$. For ease of notations, let us denote in this paragraph $\mathcal{B}_u = \mathcal{B}\left(u, C_u\left(\sqrt{n/\log n}\right)\right)$ and $\mathcal{B}_v(q) = \mathcal{B}\left(v, C_v(q)\right)$. Note that, in contrast to step (ii), we now look at the FPP up to step $\sqrt{n/\log n}$ instead of the FPP up to step $\sqrt{n \log n}$. We define

$$Q \;=\; \min\{q \in [n] \colon \mathcal{B}_u \cap \mathcal{B}_v(q) \neq \emptyset\}.$$

We have

$$
\begin{aligned}
\mathbb{P}\left(Q = q\right) &\;=\; \mathbb{P}\left(w_q \in \mathcal{B}_u \,|\, Q > q-1\right) \mathbb{P}\left(Q > q-1\right) \\
&\;=\; \sum_{\ell \in [k]} \frac{|\mathcal{B}_u \cap \Gamma_\ell|}{|\Gamma_\ell|} \mathbb{P}\left(w_q \in \Gamma_\ell \,|\, Q > q-1\right) \mathbb{P}\left(Q > q-1\right),
\end{aligned}
$$

---

[11] Recall that if $x_1 + \cdots + x_k = m$ with $x_\ell \geq 0$ then $\max_\ell x_\ell \geq k^{-1}m$.

where, as in step (ii), the second equality holds because all vertices within the same block have the same probability of being chosen at the next step of the FPP as long as the two FPP have not collided. Using $|\Gamma_\ell| \geq (1 + o(1))\pi_{\min} n$ and $|\mathcal{B}_u \cap \Gamma_\ell| \leq |\mathcal{B}_u| \leq \sqrt{n/\log n}$ leads to

$$\mathbb{P}(Q = q) \leq \frac{1}{\pi_{\min}} \frac{1 + o(1)}{\sqrt{n \log n}}.$$

Therefore,

$$\mathbb{P}(\mathcal{B}_u \cap \mathcal{B}_v \neq \emptyset) = \sum_{q=1}^{\sqrt{n/\log n}} \mathbb{P}(Q = q) \leq \frac{1 + o(1)}{\pi_{\min} \log n} = o(1).$$

This concludes the proof of Proposition 1 when $F_{ab} \sim \text{Exp}(\lambda_{ab})$. $\qquad\square$

## A.2 General case

*Proof of Proposition 1 with non-exponentially distributed costs.* In this section, the probability densities $F_{ab}$ are regular (see Assumption 2) but not necessarily exponentially distributed. We adapt the argument provided at the beginning of Section 2 of [23]. The strategy is to transform the graph $G = (V, E, c)$ generated from a wSBM$(n, \pi, F)$ into a graph $G_{\exp} = (V, E, c_{\exp})$ with the same vertices and edge set but where the costs $c_{\exp}$ are obtained from $F$ and are exponentially distributed so that $G_{\exp}$ has the same distribution as a wSBM$(n, \pi, (\text{Exp}(\lambda_{ab}))_{ab})$. We will then show that the shortest paths in $G_{\exp}$ and $G$ are the same.

Let $f_{ab}$ be the density function of the cdf $F_{ab}$, and remember that $\lambda_{ab} = F'_{ab}(0)$. Let $F_{ab}^{-1}$ be the generalized inverse function[12] of $F_{ab}$. By regularity of $F_{ab}$ (Assumption 2), we have $\lim_{t \to 0} F_{ab}(t)/t = \lambda_{ab}$ and thus $\lim_{t \to 0} F_{ab}^{-1}(t)/t = \lambda_{ab}^{-1}$. We also denote by $g_{ab}(x) = \lambda_{ab} e^{-\lambda_{ab} x}$ and by $G_{ab}(x) = 1 - e^{-\lambda_{ab} x}$ the density and cumulative distribution functions of $\text{Exp}(\lambda_{ab})$.

We first show that Proposition 1 holds for uniformly distributed edge weights, *i.e.*, when $f_{ab}(x) = \lambda_{ab} 1(x \in [0, \lambda_{ab}^{-1}])$. Denote by $F_{ab}$ the cdf associated to $f_{ab}$, and let $(z, G) \sim \text{wSBM}(n, \pi, p, F)$ where $G = (V, E, c)$. We construct the graph $G_{\exp} = (V, E, c_{\exp})$ such that for all $(w, w') \in E$ with $z_w = \ell$ and $z_{w'} = \ell'$ we have $c_{\exp}(w, w') = G_{\ell\ell'}^{-1}(\lambda_{\ell\ell'} c(w, w'))$. Note that the (unweighted) edges of $G$ and $G_{\exp}$ are the same, only the costs $c$ and $c_{\exp}$ differ. Since $\lambda_{\ell\ell'} c(w, w') \sim \text{Unif}(0, 1)$ we have $c_{\exp}(w, w') \sim \text{Exp}(\lambda_{\ell\ell'})$. In particular, $G_{\exp}$ has the same distribution of a weighted SBM whose costs are exponentially distributed, *i.e.*, $(z, G_{\exp}) \sim \text{wSBM}(n, \pi, p, G)$ with $G = (G_{ab})_{a,b}$.

Let $\mathcal{P}(u, v)$ (resp., $\mathcal{P}_{\exp}(u, v)$) be the shortest path from $u$ to $v$ in $G$ (resp., in $G_{\exp}$), and denote by $C(u, v)$ (resp., by $C_{\exp}(u, v)$) its cost. We know from Section A.1 that $C_{\exp}(u, v) = \Theta\left(\frac{\log n}{n\rho_n}\right)$ whp.

Suppose that the edge $(w, w') \in E$ belongs to $\mathcal{P}_{\exp}(u, v)$. From $c_{\exp}(w, w') \leq C_{\exp}(u, v)$ we notice that $c_{\exp}(w, w') = O(\frac{\log n}{n\rho_n})$, and hence by Assumption 1 we have $c_{\exp}(w, w') = o(1)$. Moreover, by definition of $c_{\exp}$ we have

$$\frac{c(w, w')}{c_{\exp}(w, w')} = \frac{1}{\lambda_{\ell\ell'}} \frac{G_{\ell\ell'}(c_{\exp}(w, w'))}{c_{\exp}(w, w')}.$$

Recall that $\lim_{t \to 0} G_{\ell\ell'}(t)/t = \lambda_{\ell\ell'}$. Thus, we have for any $\epsilon > 0$ and for $n$ large enough, $1 - \epsilon < \frac{c(w, w')}{c_{\exp}(w, w')} < 1 + \epsilon$. This holds for any edge $(w, w')$ belonging to $\mathcal{P}_{\exp}(u, v)$. Therefore, the sum of the costs $c(w, w')$ over $\mathcal{P}_{\exp}(u, v)$ is at most $(1 + \epsilon)C_{\exp}(u, v)$, and hence

$$C(u, v) < (1 + \epsilon)C_{\exp}(u, v). \tag{A.14}$$

Similarly, if $(w, w') \in \mathcal{P}(u, v)$, then the upper bound (A.14) together with Assumption 1 imply that $C(u, v) = o(1)$. This in turn implies $c(w, w') = o(1)$ and hence $1 - \epsilon < \frac{c(w, w')}{c_{\exp}(w, w')} < 1 + \epsilon$. Thus, summing over all edges in $\mathcal{P}(u, v)$ leads to

$$(1 - \epsilon)C_{\exp}(u, v) < C(u, v). \tag{A.15}$$

---

[12]More precisely, we define $F_{ab}^{-1}(t) = \inf\{x \in \mathbb{R} : F_{ab}(x) \geq t\}$. Because the cdf $F_{ab}$ is increasing, $F_{ab}^{-1}(t)$ is well-defined.

Combining (A.14) with (A.15) shows that Proposition 1 holds for uniformly distributed edges weights.

Finally, if the costs $c(w, w')$ are sampled from general distributions $F_{ab}$ verifying Assumption 2, then we construct the graph $G_{\text{unif}} = (V, E, c_{\text{unif}})$ where $c_{\text{unif}}(u, v) = \lambda_{z_u z_v}^{-1} F_{z_u z_v}(c(u, v))$. We have $c_{\text{unif}}(u, v) \sim \text{Unif}([0, \lambda_{z_u z_v}^{-1}])$ and we apply the previous reasoning (by replacing the exponential distributions with uniform distributions) to conclude. $\square$

# B  Proof of Sections 2.3 and 2.4

## B.1  Proof of Theorem 1

*Proof of Theorem 1.* Let $G = (V, E, c)$ be a wSBM, and let $u, v \in V$ be two arbitrary distinct vertices such that $z_u = a$ and $z_v = b$. Then, adapting the proof[13] of [39, Corollary 1], we can write

$$p_{ab}^{\text{mb}} = -\int_0^\infty c_{uv}^*(x) \log(1 - p_{ab} F_{ab}(x)) \, dx, \tag{B.1}$$

where $c_{uv}^*(x)$ is the probability density function of the weight of the shortest path between $u$ and $v$ and $F_{ab}(x) = \int_0^x f_{ab}(x) dx$ is the cumulative distribution function of the length of an edge between two vertices belonging to communities $a$ and $b$.

Proposition 1 ensures that, with high probability, the cost $C(u, v)$ is a random variable whose support is lower and upper bounded by $\frac{1+o(1)}{\tau_{\max}} \frac{\log n}{n \rho_n}$ and $\frac{1+o(1)}{\tau_{\min}} \frac{\log n}{n \rho_n}$, respectively. Its density function $c_{uv}^*(x)$ tends therefore to zero outside these two bounds, and hence by setting the $\log(\cdot)$ factor in Equation (B.1) to the lower and, respectively, the upper bound of the support of $C(u, v)$, and by integrating next the pdf $c_{uv}^*(x)$ over the whole interval, Equation (B.1) implies that

$$-\log\left(1 - p_{ab} F_{ab}\left(\frac{1+o(1)}{\tau_{\max}} \frac{\log n}{n \rho_n}\right)\right) \leq p_{ab}^{\text{mb}} \leq -\log\left(1 - p_{ab} F_{ab}\left(\frac{1+o(1)}{\tau_{\min}} \frac{\log n}{n \rho_n}\right)\right).$$

We finish the proof using $F_{ab}\left(\frac{(1+o(1))\log n}{\tau_{\min} n \rho_n}\right) = (1 + o(1))\lambda_{ab} \frac{\log n}{\tau_{\min} n \rho_n}$ and $F_{ab}\left(\frac{(1+o(1))\log n}{\tau_{\max} n \rho_n}\right) = (1 + o(1))\lambda_{ab} \frac{\log n}{\tau_{\max} n \rho_n}$ (Assumption 2). $\square$

## B.2  Proof of Theorem 2

*Proof of Theorem 2.* Let $G = (V, E, c)$ be the original graph and $G^{\text{mb}} = (V, E^{\text{mb}}, c^{\text{mb}})$ its metric backbone. Let $E^\theta \subseteq E$ be the subset of edges whose cost is no more than $\theta$:

$$(u, v) \in E^\theta \iff c(u, v) \leq \theta,$$

and which is the edge set of the corresponding threshold graph $G^\theta = (V, E^\theta, c^\theta)$, where $c^\theta$ is the restriction of $c$ to $E^\theta$.

Denote by $W, W^{\text{mb}}, W^\theta \in \mathbb{R}_+^{n \times n}$ the adjacency matrices of $G$, $G^{\text{mb}}$, and $G^\theta$, respectively.

**Overview of the proof.**  To prove Theorem 2, the key idea is to choose a threshold $\theta$ large enough such that the threshold graph contains the metric backbone (*i.e.*, $E^{\text{mb}} \subseteq E^\theta$), but not too large so that the adjacency matrices $W^{\text{mb}}$ and $W^\theta$ are not too different.

Lemma 1 ensures that, for any $\epsilon > 0$, we have $\mathbb{P}\left(\max_{1 \leq u, v \leq n} C(u, v) \leq \frac{3+\epsilon}{\tau_{\min}} \frac{\log n}{n \rho_n}\right) = 1 - o(1)$. Hence, $E^{\text{mb}} \subseteq E^\theta$ whp as soon as $\theta \geq \frac{3+\epsilon}{\tau_{\min}} \frac{\log n}{n \rho_n}$. We choose $\theta = \frac{4}{\tau_{\min}} \frac{\log n}{n \rho_n}$, and we proceed by conditioning on the event $E^{\text{mb}} \subseteq E^\theta$, which occurs with high probability given our choice of $\theta$.

We will first prove that the clusters can exactly be recovered using the eigenvectors of $\mathbb{E}W^{\text{mb}}$. Then, using Davis-Kahan's Theorem [44, Theorem 2], we show that the clusters can also be recovered from the adjacency matrix $W^{\text{mb}}$, provided that $\|W^{\text{mb}} - \mathbb{E}W^{\text{mb}}\|$ is small enough. More precisely, we obtain an upper-bound on $\text{loss}(z, \tilde{z})$ that depends on $\|W^{\text{mb}} - \mathbb{E}W^{\text{mb}}\|$. The main ingredients of the proof are thus (i) the justification of the choice of $\theta$ in Lemma 1 and (ii) the careful upper-bounding of $\|W^{\text{mb}} - \mathbb{E}W^{\text{mb}}\|$.

---

[13]We note that [39] state the result for a weighted Erdős-Rényi random graph, but their proof holds for a wSBM as well.

**Starting point: eigenstructure of the expected adjacency matrix $\mathbb{E}W^{\mathrm{mb}}$ and Davis-Kahan.** Let $Z \in \{0,1\}^{n \times k}$ be the one-hot encoding of the true community structure $z \in [k]^n$, *i.e.*,

$$\forall u \in [n],\ \forall a \in [k]\colon\ Z_{ua} \;=\; \begin{cases} 1 & \text{if } z_u = a, \\ 0 & \text{otherwise.} \end{cases} \tag{B.2}$$

For any two vertices $u, v$ belonging to communities $a$ and $b$, respectively, we write $\mathbb{E}W^{\mathrm{mb}}_{uv} = m_{ab}$. Let $M = (m_{ab}) \in \mathbb{R}^{k \times k}$. We have

$$\mathbb{E}W^{\mathrm{mb}} \;=\; ZMZ^T.$$

Denote by $|\bar\sigma_1| \geq \cdots \geq |\bar\sigma_k|$ the $k$ eigenvalues of $\mathbb{E}W^{\mathrm{mb}}$, and by $\bar u_1, \cdots, \bar u_k$ their associated eigenvectors. Let $\bar\Sigma = \operatorname{diag}(\bar\sigma_1, \cdots, \bar\sigma_k) \in \mathbb{R}^{k \times k}$ and $\bar U = [\bar u_1, \cdots, \bar u_k] \in \mathbb{R}^{n \times k}$. We have

$$\mathbb{E}W^{\mathrm{mb}} \;=\; \bar U \bar\Sigma \bar U^T. \tag{B.3}$$

Let $\Delta = \operatorname{diag}(\sqrt{n\pi}) \in \mathbb{R}^{k \times k}$ be the diagonal matrix whose diagonal elements are $\sqrt{n\pi_1}, \cdots, \sqrt{n\pi_k}$. We have

$$ZMZ^T \;=\; \left(Z\Delta^{-1}\right)\Delta M \Delta \left(Z\Delta^{-1}\right)^T.$$

Notice that $Z\Delta^{-1} \in \mathbb{R}^{n \times k}$ has orthonormal rows (indeed $\left(Z\Delta^{-1}\right)^T\left(Z\Delta^{-1}\right) = \Delta^{-1}Z^T Z \Delta^{-1} = I_K$ because $Z^T Z = \Delta^2$). Let $ODO^T$ be an eigendecomposition of the symmetric real-valued matrix $\Delta M \Delta$ (that is, $D \in \mathbb{R}^{k \times k}$ is a diagonal matrix whose diagonal elements are in decreasing order (in absolute value) and $O \in \mathbb{R}^{k \times k}$ is an orthonormal matrix). Then

$$ZMZ^T \;=\; \left(Z\Delta^{-1}O\right)D\left(Z\Delta^{-1}O\right)^T$$

is an eigendecomposition of $ZMZ^T$ (because $Z\Delta^{-1}O$ is orthonormal). Hence, going back to (B.3), we have $\bar\Sigma = D$ and $\bar U = Z\Delta^{-1}O$ for some orthonormal matrix $O \in \mathbb{R}^{k \times k}$.

Because $\bar U = Z\Delta^{-1}O$, two vertices are in the same cluster if and only if their corresponding rows in $\bar U$ are the same. In other words, the spectral embedding of the expected graph $\mathbb{E}W^{\mathrm{mb}}$ is condensed into $k$ points $(\Delta^{-1}O)_{1\cdot}, \cdots, (\Delta^{-1}O)_{k\cdot}$ of $\mathbb{R}^k$. Consequently, $k$-means on $\bar U$ recovers the true clusters (up to a permutation).

Next, [27, Lemma 5.3] ensures that any $(1+\epsilon)$ solution $\tilde z$ of the $k$-means problem on $U\Sigma$ verifies

$$\operatorname{loss}(z, \tilde z) \;\leq\; 4(4 + 2\epsilon) \min_{O \in \mathbf{O}_{k \times k}} \|UO - \bar U\|_F^2 \tag{B.4}$$

where $\mathbf{O}_{k \times k}$ denotes the group of orthonormal $k$-by-$k$ matrices and $\|\cdot\|_F$ is the Frobenius norm. Davis-Kahan's Theorem [44, Theorem 2] ensures the existence of an orthogonal matrix $O \in \mathbf{O}_{k \times k}$ such that

$$\|UO - \bar U\|_F \;\leq\; 2^{3/2}k^{1/2}\frac{\|W^{\mathrm{mb}} - \mathbb{E}W^{\mathrm{mb}}\|}{|\bar\sigma_k|}, \tag{B.5}$$

where $\|\cdot\|$ denotes the matrix operator norm.

Let us now establish an expression of $|\bar\sigma_k|$. Observe firstly that $\Delta M \Delta$ and $M\Delta^2$ have the same eigenvalues.[14] From Lemma 4, we have

$$\frac{1}{2\tau_{\max}^2}(\Lambda \odot B)_{ab}\frac{\log^2 n}{n^2\rho_n} \;\leq\; m_{ab} \;\leq\; \frac{1}{2\tau_{\min}^2}(\Lambda \odot B)_{ab}\frac{\log^2 n}{n^2\rho_n}.$$

The definition of $T = [\Lambda \odot B]\operatorname{diag}(\pi)$ in (2.1) and the fact that $\Delta^2 = n\operatorname{diag}(\pi)$ further imply that

$$\frac{1}{2\tau_{\max}^2}T_{ab}\frac{\log^2 n}{n\rho_n} \;\leq\; \left(M\Delta^2\right)_{ab} \;\leq\; \frac{1}{2\tau_{\min}^2}T_{ab}\frac{\log^2 n}{n\rho_n}.$$

Recall that $\theta = \frac{4}{\tau_{\min}}\frac{\log n}{n\rho_n}$ and $\mu$ is the smallest (in absolute value) eigenvalue of $T$. Using the assumption $\tau_{\min} = \tau_{\max}$, we obtain that $M\Delta^2 = \frac{\theta \log n}{8\tau_{\min}}T$ and thus

$$\bar\sigma_k \;=\; \frac{\mu}{8\tau_{\min}}\theta \log n. \tag{B.6}$$

Therefore, combining (B.4), (B.5) and (B.6), we obtain

$$\operatorname{loss}(z, \tilde z) \;\leq\; (4 + 2\epsilon)2^5 k \left(\frac{8\tau_{\min}}{\mu} \cdot \frac{\|W^{\mathrm{mb}} - \mathbb{E}W^{\mathrm{mb}}\|}{\theta \log n}\right)^2. \tag{B.7}$$

---

[14]Let $A, B$ be two symmetric matrices of the same size. Let $\lambda$ be an eigenvalue of $ABA$, with corresponding eigenvector $x$. Multiplying both sides of $ABAx = \lambda x$ by $A$, we get $A^2 By = \lambda y$ with $y = Ax$.

**Core of the proof: concentration of $W^{\mathrm{mb}}$ around $\mathbb{E}W^{\mathrm{mb}}$.** We finish the proof by showing that $\|W^{\mathrm{mb}} - \mathbb{E}W^{\mathrm{mb}}\| = O\left(\theta\sqrt{\log n}\right)$ whp. First, by a triangle inequality, we have

$$\|W^{\mathrm{mb}} - \mathbb{E}W^{\mathrm{mb}}\| \;\leq\; \|W^{\mathrm{mb}} - W^\theta - \mathbb{E}\left[W^{\mathrm{mb}} - W^\theta\right]\| + \|W^\theta - \mathbb{E}W^\theta\| \tag{B.8}$$

For ease of the exposition, we will upper-bound the term of (B.8) in the following order: (i) the second term $\|W^\theta - \mathbb{E}W^\theta\|$, and then (ii) the first term $\|W^{\mathrm{mb}} - W^\theta - \mathbb{E}\left[W^{\mathrm{mb}} - W^\theta\right]\|$.

(i) Let us first study $\|W^\theta - \mathbb{E}W^\theta\|$. The matrix $X = W^\theta/\theta$ is symmetric with zero-diagonal, whose entries $\{X_{uv}, u < v\}$ are independent $[0,1]$-valued random variables. Moreover, Lemma 3 shows that $\mathbb{E}\left[X_{uv} \mid z_u = a, z_v = b\right] = p_{ab}\lambda_{ab}\theta = \Theta(\log n/n)$. Thus, [18, Theorem 5] ensures that for any $c > 0$ there exists $c' > 0$ such that

$$\mathbb{P}\left(\|W^\theta - \mathbb{E}W^\theta\| \geq c'\theta\sqrt{\log n}\right) \;\leq\; n^{-c}. \tag{B.9}$$

(ii) Next, let us study $\|W^{\mathrm{mb}} - W^\theta - \mathbb{E}\left[W^{\mathrm{mb}} - W^\theta\right]\|_2$. Denote $Y = -\left(W^{\mathrm{mb}} - W^\theta\right)/\theta$. By Lemma 1 and our choice of $\theta$, we have $E^{\mathrm{mb}} \subseteq E^\theta$ (for $n$ large enough). Moreover, $W^\theta_{uv} = W^{\mathrm{mb}}_{uv}$ for all $\{u,v\} \in E^{\mathrm{mb}} \cap E^\theta = E^{\mathrm{mb}}$. Hence,

$$Y_{uv} \;=\; \begin{cases} \frac{W^\theta_{uv}}{\theta} & \text{if } \{u,v\} \in E^\theta \backslash E^{\mathrm{mb}}, \\ 0 & \text{otherwise,} \end{cases} \quad \text{and} \quad \mathbb{E}Y_{uv} \;=\; \begin{cases} \frac{\mathbb{E}W^\theta_{uv}}{\theta} & \text{if } \{u,v\} \in E^\theta \backslash E^{\mathrm{mb}}, \\ 0 & \text{otherwise.} \end{cases}$$

We can thus rewrite the matrices $Y$ and $\mathbb{E}Y$ as $Y = R \odot W^\theta/\theta$ and $\mathbb{E}Y = R \odot \mathbb{E}W^\theta/\theta$, where $\odot$ denote the Hadamard (entry-wise) matrix product and $R \in \{0,1\}^{n\times n}$ is defined by

$$R_{uv} \;=\; R_{vu} \;=\; \begin{cases} 1 & \text{if } \{u,v\} \in E^\theta \backslash E^{\mathrm{mb}}, \\ 0 & \text{otherwise.} \end{cases}$$

Let $\mathcal{E}_{c_1}$ be the event that all rows of $R$ have at most $c_1 \log n$ non-zero entries (with $c_1 > 0$), *i.e.,*

$$\mathcal{E}_{c_1} \;=\; \left\{ \forall u \in [n]\colon \sum_{v=1}^n \mathbb{1}\{R_{uv} \neq 0\} \;\leq\; c_1 \log n \right\}.$$

Because $E^{\mathrm{mb}} \subseteq E^\theta$, we have

$$\mathbb{1}\{R_{uv} \neq 0\} \;=\; R_{uv} \;=\; \mathbb{1}\{(u,v) \in E^\theta \backslash E^{\mathrm{mb}}\} \;\leq\; \mathbb{1}\{(u,v) \in E^\theta\}.$$

Thus, $\mathcal{E}_{c_1,\theta} \subseteq \mathcal{E}_{c_1}$, where

$$\mathcal{E}_{c_1,\theta} \;=\; \left\{ \forall u \in [n]\colon \sum_{v=1}^n \mathbb{1}\{(u,v) \in E^\theta\} \;\leq\; c_1 \log n \right\}.$$

Hence, $\mathbb{P}(\mathcal{E}_{c_1}) \geq \mathbb{P}(\mathcal{E}_{c_1,\theta})$. Recall also that $\mathbb{1}\{(u,v) \in E^\theta\} = \mathbb{1}\{c(u,v) \leq \theta\}$. Thus, by Lemma 2, for any $c_0 > 0$ there exists a $c_1 > 0$ such that

$$\mathbb{P}\left(\mathcal{E}_{c_1}\right) \;\geq\; 1 - n^{-c_0}. \tag{B.10}$$

Conditioned on the high probability event $\mathcal{E}_{c_1}$, every row of the matrix $R$ has at most $c_1 \log n$ non-zero elements. Moreover, $W/\theta$ is symmetric and has bounded (hence sub-gaussian) entries. Therefore [5, Corollary 3.9] ensures the existence of constants $C, C' > 0$ such that

$$\mathbb{P}\left(R \odot \left(W^\theta - \mathbb{E}W^\theta\right)/\theta \;\geq\; C\sqrt{\log n} + t \,\Big|\, \mathcal{E}_{c_1}\right) \;\leq\; e^{-C't^2}.$$

Using $t = C''\sqrt{\log n}$, and because $Y - \mathbb{E}Y = R \odot \left(W^\theta - \mathbb{E}W^\theta\right)/\theta$, we obtain the following statement: for any $c > 0$, there exists $c' > 0$ such that

$$\mathbb{P}\left(\|Y - \mathbb{E}Y\|_2 \;\geq\; c'\sqrt{\log n} \,\Big|\, \mathcal{E}_{c_1}\right) \;\leq\; n^{-c}.$$

Using (B.10), we finally obtain

$$\mathbb{P}\left(\|W^{\mathrm{mb}} - W^\theta - \mathbb{E}\left[W^{\mathrm{mb}} - W^\theta\right]\| \;\geq\; c'\theta\sqrt{\log n}\right) \;\leq\; 2n^{-c}. \tag{B.11}$$

**Conclusion** Using (B.8), (B.9) and (B.11), we have

$$\|W^{\mathrm{mb}} - \mathbb{E}W^{\mathrm{mb}}\| \leq 2c'\theta\sqrt{\log n}, \tag{B.12}$$

with probability at least $1 - 3n^{-c}$. The proof ends by combining (B.12) with (B.7). $\qquad\square$

## B.3 Additional Lemmas

**Lemma 1.** *Let* $(z, G) \sim \mathrm{wSBM}(n, \pi, p, F)$ *and suppose that Assumptions 1 and 2 hold. Then,*

$$\max_{u,v\in[n]} C(u,v) \leq (1 + o(1))\frac{3}{\tau_{\min}}\frac{\log n}{n\rho_n}.$$

*Proof.* We proceed as in the proof of Proposition 1 by firstly assuming that $F_{ab} \sim \mathrm{Exp}(\lambda_{ab})$. The extension to more general weight distributions follows from the same coupling argument presented in Section A.2 and is thus omitted.

We use the same notations as in the proof of Proposition 1. Recall in particular the notations $C_u(q)$, $U_\ell(q)$ and $X_{\ell\ell'}(u, q)$. Because we established that $\mathcal{B}(u, C_u(\sqrt{n\log n})) \cap \mathcal{B}(v, C_v(\sqrt{n\log n})) \neq \emptyset$ whp, we have

$$C(u,v) \leq C_u(\sqrt{n\log n}) + C_v(\sqrt{n\log n}).$$

Therefore,

$$\max_{u,v} C(u,v) \leq \max_{u,v}\left(C_u(\sqrt{n\log n}) + C_v(\sqrt{n\log n})\right)$$
$$= 2\max_u C_u(\sqrt{n\log n}).$$

Using a union bound, we obtain for any $t > 0$,

$$\mathbb{P}\left(\max_u C_u(\sqrt{n\log n}) \geq t\right) \leq \sum_{u=1}^n \mathbb{P}\left(C_u(\sqrt{n\log n}) \geq t\right)$$
$$= n\mathbb{P}\left(C_{u^*}(\sqrt{n\log n}) \geq t\right), \tag{B.13}$$

where $u^*$ denotes an arbitrary vertex chosen in $[n]$. For any $s > 0$, we have (by Chernoff bounds)

$$\mathbb{P}\left(C_{u^*}(\sqrt{n\log n}) \geq t\right) \leq e^{-st}\mathbb{E}\left[e^{sC_{u^*}(\sqrt{n\log n})}\right]. \tag{B.14}$$

Recall from (A.3) that $C_{u^*}(q+1) - C_{u^*}(q) \mid S(u, \cdot)$ are i.i.d. $\mathrm{Exp}(\theta_q)$ with $\theta_q = \sum_{\ell,\ell'} \lambda_{\ell\ell'} S_{\ell\ell'}(u, q)$. As for $X \sim \mathrm{Exp}(\theta)$ we have $\mathbb{E}[e^{sX}] = \theta/(\theta - s) = 1 + s/(\theta - s)$,

$$\mathbb{E}\left[e^{sC_{u^*}(\sqrt{n\log n})} \mid S(u, \cdot)\right] = \prod_{q=1}^{\sqrt{n\log n}} \mathbb{E}\left[e^{s(C_{u^*}(q+1) - C_{u^*}(q))} \mid S(u, \cdot)\right]$$
$$= \prod_{q=1}^{\sqrt{n\log n}}\left(1 + \frac{s}{\theta_q - s}\right)$$
$$= \exp\left(\sum_{q=1}^{\sqrt{n\log n}} \log\left(1 + \frac{s}{\theta_q - s}\right)\right)$$
$$\leq \exp\left(\sum_{q=1}^{\sqrt{n\log n}} \frac{s}{\theta_q - s}\right).$$

Let $0 < \epsilon < 1/6$. From Equations (A.5) and (A.6), we have whp (for $n$ large enough)

$$\theta_q \geq (1 - \epsilon)n\rho_n\tau_{\min}q.$$

Choose $s = (1 - 2\epsilon)n\rho_n\tau_{\min}$ and let $\alpha_n$ be a diverging sequence verifying $\alpha_n = o(\sqrt{n\log n})$ to be chosen later. We split the sum as follows

$$\sum_{q=1}^{\sqrt{n\log n}} \frac{s}{\theta_q - s} = \sum_{q=1}^{\alpha_n} \frac{s}{\theta_q - s} + \sum_{q=\alpha_n+1}^{\sqrt{n\log n}} \frac{s}{\theta_q - s}.$$

We have for any $1 \le q \le \alpha_n$

$$\theta_q - s \ge n\rho_n\tau_{\min}q\left(1 - \epsilon - \frac{1-2\epsilon}{q}\right) \ge n\rho_n\tau_{\min}q\epsilon,$$

while for $\alpha_n \ge q$ we have $(1-2\epsilon)\alpha_n \le \epsilon$ (for $n$ large enough) and thus

$$\theta_q - s \ge n\rho_n\tau_{\min}q\left(1 - \epsilon - \frac{1-2\epsilon}{q}\right) \ge n\rho_n\tau_{\min}q\left(1 - \epsilon - \frac{1-2\epsilon}{\alpha_n}\right) \ge n\rho_n\tau_{\min}q(1-2\epsilon),$$

and thus

$$\sum_{q=1}^{\sqrt{n\log n}} \frac{s}{\theta_q - s} \le (1-2\epsilon)\left[\frac{1}{\epsilon}\sum_{q=1}^{\alpha_n}\frac{1}{q} + \frac{1}{1-2\epsilon}\sum_{q=\alpha_n+1}^{\sqrt{n\log n}}\frac{1}{q}\right].$$

Recalling that $\sum_{q=m+1}^{n} q^{-1} \le \log(n/m)$ for any $m \ge 1$ (by integration) further leads to

$$\sum_{q=1}^{\sqrt{n\log n}} \frac{s}{\theta_q - s} \le (1-2\epsilon)\left[\frac{1}{\epsilon}(1 + \log\alpha_n) + \frac{1}{1-2\epsilon}\log\left(\frac{\sqrt{n\log n}}{\alpha_n}\right)\right].$$

Therefore,

$$\mathbb{E}\left[e^{sC_{u^*}(\sqrt{n\log n})} \mid S(u, \cdot)\right] \le \exp\left((1-2\epsilon)\left[\frac{1}{\epsilon}(1 + \log\alpha_n) + \frac{1}{1-2\epsilon}\log\left(\frac{\sqrt{n\log n}}{\alpha_n}\right)\right]\right).$$

Because the right-hand side of this last upper bound does not depend on $S(u, \cdot)$, we have by the total law of probabilities,

$$\mathbb{E}\left[e^{sC_{u^*}(\sqrt{n\log n})}\right] \le \exp\left((1-2\epsilon)\left[\frac{1}{\epsilon}(1 + \log\alpha_n) + \frac{1}{1-2\epsilon}\log\left(\frac{\sqrt{n\log n}}{\alpha_n}\right)\right]\right). \quad \text{(B.15)}$$

We choose $\alpha_n = \lceil\sqrt{\log n}\rceil$. Because $\alpha_n = \omega(1)$, $\alpha_n = o(\log n)$, and $\epsilon$ is fixed, we have for $n$ large enough,

$$\frac{1}{\epsilon}(1 + \log\alpha_n) \le \epsilon\frac{\log n}{2}$$
$$\frac{1}{1-2\epsilon}\log\left(\frac{\sqrt{n\log n}}{\alpha_n}\right) \le (1+3\epsilon)\frac{\log n}{2},$$

where the second inequality uses $\epsilon < 1/6$. Thus, we can rewrite the inequality (B.15) (for $n$ large enough) as

$$\mathbb{E}\left[e^{sC_{u^*}(\sqrt{n\log n})}\right] \le \exp\left((1-2\epsilon)(1+4\epsilon)\frac{\log n}{2}\right) \le \exp\left((1+2\epsilon)\frac{\log n}{2}\right).$$

Let $t = (3 + \epsilon')\log n/(2n\rho_n\tau_{\min})$ with $\epsilon' = 15\epsilon$. Recall that $s = (1 - 2\epsilon)n\rho_n\tau_{\min}$. From (B.14) we obtain

$$\mathbb{P}\left(C_{u^*}(\sqrt{n\log n}) \ge t\right) \le e^{-(1-2\epsilon)(3+\epsilon')\frac{\log n}{2} + (1+2\epsilon)\frac{\log n}{2}}$$
$$= e^{-\frac{\log n}{2}\left(2 - 8\epsilon - 2\epsilon\epsilon' + \epsilon'\right)}$$
$$\le e^{-\log n(1+\epsilon)},$$

where in the last line we uses $-8\epsilon - 2\epsilon\epsilon' + \epsilon' = 6\epsilon - 30\epsilon^2 \ge \epsilon$ (using $\epsilon < 1/6$). Because $1 + \epsilon > 1$, we have $\mathbb{P}\left(C_{u^*}(\sqrt{n\log n}) \ge t\right) = o(n^{-1})$, and we conclude using (B.13). $\qquad\square$

**Lemma 2.** *Let $(z, G) \sim \text{wSBM}(n, \pi, p, F)$ and suppose Assumptions 1 and 2 hold. Let $c_0, c_1 > 0$ be two constants (independent of $n$), and let $\theta = c_0 \frac{\log n}{n \rho_n}$. Denote*

$$\mathcal{E}_{c_1, \theta} \;=\; \left\{ \forall u \in [n] \colon \sum_{v=1}^{n} \mathbb{1}\{c(u, v) \leq \theta\} \;\leq\; (4 B_{\max} \lambda_{\max} c_0 + c_1) \log n \right\}.$$

*Then, $\mathbb{P}\left(\mathcal{E}_{c_1, \theta}\right) \;\geq\; 1 - n^{-c_1}$.*

*Proof.* For $u \neq v \in [n]$ such that $z_u = a$ and $z_v = b$, we have (recall that the probability of having an edge $(u, v) \in E$ is $p_{ab}$, and the cost of this edge is sampled from $F_{ab}$)

$$\mathbb{P}\left(c(u, v) \leq \theta \mid z_u = a, z_v = b\right) \;=\; p_{ab} F_{ab}(\theta).$$

Recall also that, by Assumption 1, we have $p_{ab} = B_{ab} \rho_n$ and by Assumption 1 $F_{ab}(x) = (1 + o(1)) \lambda_{ab} x$. Let $\epsilon > 0$. Using the law of total probability, we obtain

$$\mathbb{P}\left(c(u, v) \leq \theta\right) \;\leq\; (1 + o(1)) B_{\max} \lambda_{\max} \theta \rho_n,$$

where $B_{\max} = \max_{1 \leq a, b \leq k} B_{ab}$ and $\lambda_{\max} = \max_{1 \leq a, b \leq k} \lambda_{ab}$. Thus, for $n$ large enough we have

$$\mathbb{P}\left(c(u, v) \leq \theta\right) \;\leq\; 2 B_{\max} \lambda_{\max} \theta \rho_n,$$

For ease of notations, let $p_\theta = 2 B_{\max} \lambda_{\max} \theta \rho_n$ and $\tilde{c} = 4 B_{\max} \lambda_{\max} c_0 + 1 + c_1$.

For $u \in [n]$ we denote $d_u = \sum_{v=1}^{n} \mathbb{1}\{c(u, v) \leq \theta\}$. We have

$$\mathbb{P}\left(\max_{u \in [n]} d_u \geq \tilde{c} \log n\right) \leq \sum_{u \in [n]} \mathbb{P}\left(d_u \geq \tilde{c} \log n\right). \tag{B.16}$$

Moreover, for any $t > 0$, we have

$$\mathbb{E}\left[e^{t d_u}\right] \;\leq\; \left(1 - p_\theta + p_\theta e^t\right)^n \;\leq\; e^{n p_\theta \left(e^t - 1\right)},$$

where the second inequality used $\log(1 + x) \leq x$. Let $\tilde{c} > 0$. Using Chernoff's bounds, we have

$$\mathbb{P}\left(d_u \geq \tilde{c} \log n\right) \;\leq\; e^{-t \tilde{c} \log n + n p_\theta (e^t - 1)}.$$

Let $t = 1$. Because $e^1 - 1 \leq 2$ and $n p_\theta = 2 B_{\max} \lambda_{\max} c_0 \log n$, we obtain

$$\mathbb{P}\left(d_u \geq \tilde{c} \log n\right) \;\leq\; e^{-\log n (\tilde{c} \log n - 4 B_{\max} \lambda_{\max} c_0)}.$$

We finish the proof by letting $\tilde{c} = 4 B_{\max} \lambda_{\max} c_0 + 1 + c_1$ and using (B.16). $\qquad \square$

**Lemma 3.** *Let $(z, G) \sim \text{wSBM}(n, \pi, p, F)$ and suppose Assumptions 1 and 2 hold. Let $\theta = c \frac{\log n}{n \rho_n}$. Denote $W$ the weighted adjacency matrix of $G$, and $W^\theta$ the weighted adjacency matrix of the threshold graph. Let $u, v \in [n]$ such that $z_u = a$ and $z_v = b$. We have $\mathbb{E}\left[W_{uv}^\theta\right] = p_{ab} \lambda_{ab} \theta^2 / 2$.*

*Proof.* We have $W_{uv}^\theta = W_{uv} \mathbb{1}\{W_{uv} \leq \theta\}$. Moreover, $W_{uv} \mid W_{uv} \neq 0$ is sampled from $F_{ab}$. Thus,

$$\mathbb{E}\left[W_{uv}^\theta\right] \;=\; p_{ab} \mathbb{E}\left[W_{uv} \mathbb{1}\{W_{uv} \leq \theta\}\right].$$

Denote also $f_{ab}$ the pdf of $F_{ab}$. Recall by Assumption 2 that $f_{ab}(x) = \lambda_{ab} + O(x)$. Because $\theta \ll 1$ we have

$$\mathbb{E}\left[W_{uv}^\theta\right] \;=\; p_{ab} \lambda_{ab} \frac{\theta^2}{2}.$$

$$\square$$

**Lemma 4.** *Let $(z, G) \sim \text{wSBM}(n, \pi, p, F)$ and suppose Assumptions 1 and 2 hold. Let $\theta = c \frac{\log n}{n \rho_n}$. Denote $W$ the weighted adjacency matrix of $G$, and $W^{\text{mb}}$ the weighted adjacency matrix of the metric backbone of $G$. Let $u, v \in [n]$ such that $z_u = a$ and $z_v = b$. We have*

$$\frac{1}{2 \tau_{\max}^2} (\Lambda \odot B)_{ab} \frac{\log^2 n}{n^2 \rho_n} \;\leq\; \mathbb{E}\left[W_{uv}^{\text{mb}}\right] \;\leq\; \frac{1}{2 \tau_{\min}^2} (\Lambda \odot B)_{ab} \frac{\log^2 n}{n^2 \rho_n}.$$

*Proof.* Let $u, v \in [n]$ be two arbitrary vertices such that $z_u = a$ and $z_v = b$. We have $W_{uv}^{\mathrm{mb}} = c(u,v)\mathbb{1}\{(u,v) \in E^{\mathrm{mb}}\}$. Recall from Proposition 1 that

$$(\tau_{\max})^{-1} \leq \frac{n\rho_n}{\log n}C(u,v) \leq (\tau_{\min})^{-1}$$

whp, where $C(u,v)$ is the cost of the shortest path from $u$ to $v$. Therefore,

$$
\begin{aligned}
\mathbb{E}\left[W_{uv}^{\mathrm{mb}}\right] &= \mathbb{E}\left[c(u,v) \cap (u,v) \in E^{\mathrm{mb}}\right] \\
&\leq \mathbb{E}\left[c(u,v)\mathbb{1}\left\{c(u,v) \leq \frac{1}{\tau_{\min}}\frac{\log n}{n\rho_n}\right\}\right] \\
&= p_{ab}\lambda_{ab}\frac{1}{2}\left(\frac{\log n}{\tau_{\min}n\rho_n}\right)^2,
\end{aligned}
$$

where we used Lemma 3 with $\theta = \frac{1}{\tau_{\min}}\frac{\log n}{n\rho_n}$. Similarly,

$$
\begin{aligned}
\mathbb{E}\left[W_{uv}^{\mathrm{mb}}\right] &\geq \mathbb{E}\left[c(u,v)\mathbb{1}\left\{c(u,v) \leq \frac{1}{\tau_{\max}}\frac{\log n}{n\rho_n}\right\}\right] \\
&= p_{ab}\lambda_{ab}\frac{1}{2}\left(\frac{\log n}{\tau_{\max}n\rho_n}\right)^2.
\end{aligned}
$$

Recalling that $p_{ab} = B_{ab}\rho_n$, we obtain

$$\frac{1}{2\tau_{\max}^2}(\Lambda \odot B)_{ab}\frac{\log^2 n}{n^2\rho_n} \leq \mathbb{E}\left[W_{uv}^{\mathrm{mb}}\right] \leq \frac{1}{2\tau_{\min}^2}(\Lambda \odot B)_{ab}\frac{\log^2 n}{n^2\rho_n}.$$

$\square$

## C  Additional Numerical Experiments

### C.1  Additional Material for Section 4

| Data set | $n$ | $|E|$ | $k$ | $\bar{d}$ |
|---|---|---|---|---|
| High school | 327 | 5,818 | 9 | 36 |
| Primary school | 242 | 8,317 | 11 | 69 |
| DBLP | 3,762 | 17,587 | 193 | 9 |
| Amazon | 8,035 | 183,663 | 195 | 46 |

Table 2: Dimensions of networks considered. *High school* and *primary school* data sets are taken from http://www.sociopatterns.org, where the weights represent the number of interactions between students. *DBLP* and *Amazon* data sets are unweighted and taken from https://snap.stanford.edu/data/. $\bar{d}$ denotes the average unweighted degree, *i.e.,* $2|E|/n$.

| Data set | High school | Primary school | DBLP | Amazon |
|---|---|---|---|---|
| $\frac{|E^{\mathrm{mb}}|}{|E|}$ | 0.29 | 0.34 | 0.82 | 0.72 |

Table 3: Ratio of edges kept by the *metric backbone* in various real graphs.

We highlight the difference between the metric backbone and the spectral sparsification of the *Primary school* data set in Figure 5. Whereas Figure 2 highlights a clear pattern between the metric backbone and the threshold graph, finding from Figure 5 any pattern in the edges present in the metric backbone but not in the spectral sparsification (and vice-versa) is much harder. Although both sparsified graphs preserve the community structure very well, they do so by keeping a very different set of edges.

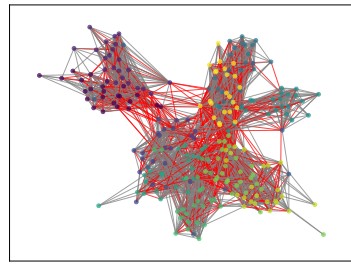

(a) *Metric Backbone*

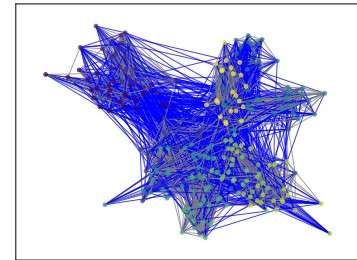

(b) *Spectral Sparsifier*

Figure 5: Graphs obtained from *Primary school* data set, after taking the metric backbone (Figure 5a) and after *spectral sparsification* (Figure 5b), drawn using the same layout. Vertex colors represent the true clusters. Edges present in the metric backbone but not in the spectral sparsifier graph are highlighted in red. Similarly, edges present in the spectral sparsifier graph, but not in the metric backbone, are highlighted in blue.

We preprocessed the DBLP and Amazon datasets by only keeping components where the second eigenvalue of the normalized Laplacian of each component that is kept is larger than 0.1. This ensures that we do not consider communities that are not well-connected.

We additionally show in Figure 6 the performance of the three clustering algorithms (Bayesian, Leiden, and spectral clustering) on four other data sets. As we do not have any ground truth for these additional data sets, we computed the ARI between the clustering obtained on the original network and on the sparsified network. We observe that the largest ARI is almost always obtained for the metric backbone. This shows that the metric backbone is the sparsification that best preserves the community structures of the original networks.

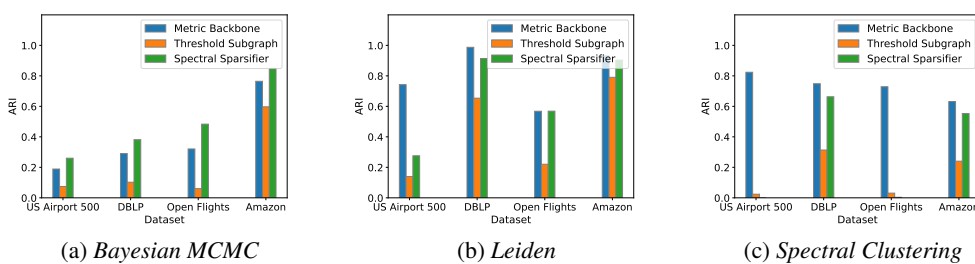

(a) *Bayesian MCMC*      (b) *Leiden*      (c) *Spectral Clustering*

Figure 6: Effect of sparsification on the performance of clustering algorithms compared to the results of the same clustering algorithms ran on the original graph.

## C.2 Additional Material for Section 5

**Number of edges in the $q$-nearest neighbor graph and its metric backbone**    We show in Figure 7 the number of edges in $G_q$ and $G_q^{\mathrm{mb}}$ (we do not show the number of edges of the spectral sparsified $G_q^{\mathrm{ss}}$ because the hyperparameters of $G_q^{\mathrm{ss}}$ are chosen such that $G_q^{\mathrm{mb}}$ and $G_q^{\mathrm{ss}}$ have the same number of edges).

**Exploring another similarity measure**    Finally, we provide additional evidence supporting the robustness of the metric backbone with respect to the number of nearest neighbors $q$ by providing the results of another clustering algorithm, namely *threshold-based subspace clustering* (TSC). *TSC* is a subspace clustering algorithm and was shown to succeed under very general conditions on the high-dimensional data set to be clustered [21]. TSC performs spectral clustering on a $q$-nearest neighbors graph obtained using the following similarity measure

$$\mathrm{sim}(x_u, x_v) = \exp\left(-2\arccos\left(<x_u, x_v>\right)\right).$$

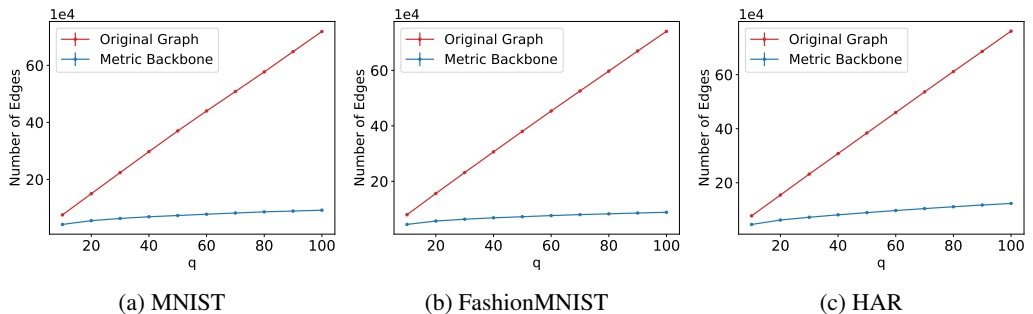

(a) MNIST            (b) FashionMNIST            (c) HAR

Figure 7: Number of edges in the $q$-nearest neighbour graph (built using Gaussian kernel similarity) and its metric backbone.

Because in Section 5 we showed the performance of spectral clustering using the Gaussian kernel similarity, these additional experiments also show the robustness of our results with respect to the choice of the similarity measure. Figure 8 shows the performance of TSC on $G_q$ and its sparsified versions $G_q^{\mathrm{mb}}$ and $G_q^{\mathrm{ss}}$, and Figure 9 shows the number of edges (before and after sparsification).

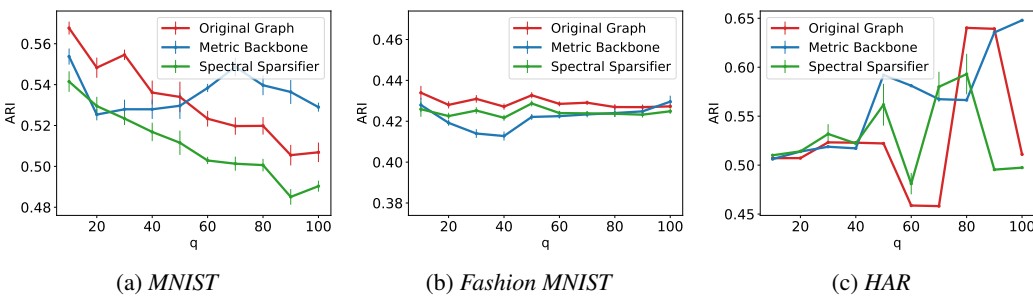

(a) *MNIST*            (b) *Fashion MNIST*            (c) *HAR*

Figure 8: Performance of TSC on subsets of MNIST and FashionMNIST datasets, and on the HAR dataset. The plots show the ARI averaged over 10 trials, and the error bars show the standard error.

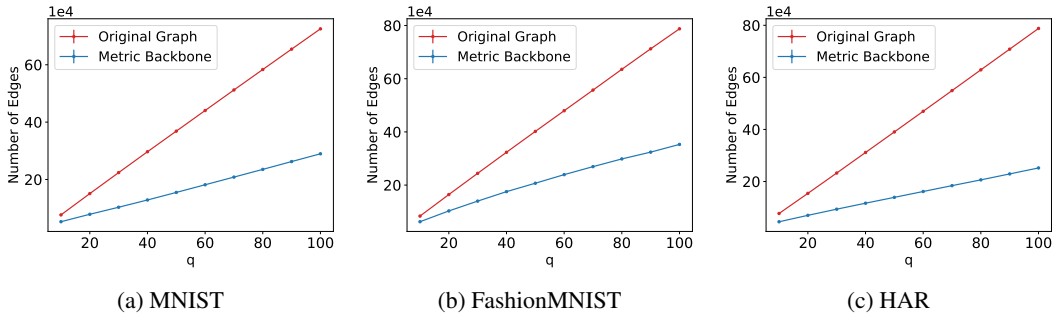

(a) MNIST            (b) FashionMNIST            (c) HAR

Figure 9: Number of edges remaining in each graph, using dot product similarity.

**Computing Resources**    All experiments were run on a standard laptop with 16GB of RAM and 12th Gen Intel(R) Core(TM) i7-1250U CPU.

To compute the full metric backbone, we used the `igraph` distances implementation [11]. It takes around 7 minutes for graphs with $10,000$ vertices and $q = 10$. For the same graph, computing the spectral sparsification takes around 15 minutes. In general, computing the spectral sparsification was 2 to 3 times slower than computing the metric backbone. Computing the approximate metric backbone takes around 100 seconds for $70,000$ vertices and $q = 132$ on our `C++` implementation.

For spectral clustering and TSC algorithms, the bottleneck is the clustering part: running the *scikit-learn* implementation of spectral clustering takes 3.5 hours for MNIST and 2 hours for Fashion-MNIST while we only needed 100 seconds to obtain the metric backbone approximation.

