# OpenReview forum: "Why the Metric Backbone Preserves Community Structure"
_NeurIPS.cc/2024/Conference — NeurIPS 2024 poster_

### Official Review · Reviewer_bM2e · 2024-07-08

**Soundness:** 3
**Presentation:** 3
**Contribution:** 3
**Rating:** 7
**Confidence:** 3

**Summary:**

For weighted distance graphs, the metric backbone corresponds to the union of the shortest paths. This work shows that the community structure of a graph can still be detected from this backbone. This is surprising, since inter-community edges often serve as bridges between communities, so one would expect them to be overrepresented in the metric backbone.
Experiments are performed to demonstrate this behavior on real-world networks.

**Strengths:**

The paper is well-written and easy to read.

The problem is interesting and the described theoretical behavior is surprising.

The overview given in Section 3.2 is helpful.

**Weaknesses:**

The experimental setup in Section 4 seems somewhat overcomplicated. Is this a standard method for converting weighted networks to distance networks? If so, refer to works where this methodology is studied.
Also, it is worth pointing out that if the proximity graph is generated from some weighted SBM, then this distance-conversion will introduce dependencies between the edge weights.

Similarly, the setup of Section 5 is also quite complicated without providing much motivation or justification. The obvious way of converting points to a distance network would be to consider the complete graph with weights corresponding to the Euclidean distance, but I understand that then every edge belongs to the metric backbone. Please explain why this complicated construction is needed.
Also, the label placement of Figure 2a is a bit inconvenient as it blocks the blue line.

Moreover, while the results of Section 4 are interesting, the results of Section 5 are not convincing.

Minor comments:
* In the denominator of $p(u,v)$ given below line 251, you write $u\in\text{Nei}(u)\wedge\dots$ instead of $u\in\text{Nei}(u)\wedge\dots$.

**Questions:**

In line 132, you write "Or as the uniform distribution", but this seems like an odd and unnecessary statement. It makes sense that you want to point out the resemblance to the exponential distribution for taking minima, but it's unclear how the resemblance to the uniform is necessary.

At first, I was a bit confused with the fact that the inequalities given in Proposition 2.1 do not depend on a or b. Perhaps you could comment on that in the text. For example, you could comment on the fact that there will always be pairs whose distances fall below this interval, but that their fraction vanishes, and that one expects intra-community distances to be smaller.

**Limitations:**

The authors have addressed the limitations of the results.

---

> ### Author Rebuttal · Authors · 2024-08-05
>
> Dear Reviewer bM2e,
>
> We thank you for your time in evaluating our submission and we are grateful for your comments. Please find below responses to the questions raised in your reviews.
>
>  * The methodology used in Section 4 is relatively standard, as we took it from previous papers on the metric backbone (see references [8] and [36] for example). We acknowledge that these transformations involve two non-linearities (the first one to transform the unweighted graph into a weighted graph, and the second one to transform the weights into distances). The theoretical Section 2 does not involve those transformations (as we consider weighted SBM where the weights are distances).
>
> * Using a Gaussian kernel to compute similarities between pairs of points is also standard. For example, this is exactly what is implemented for spectral clustering in scikit-learn (when one does not provide an adjacency matrix but a set $n$ points in $\mathbb R^d$).
>
> * Indeed, the result of Proposition 2.1 does not depend on $a$ nor $b$. This is because we upper and lower-bound the cost of the shortest path between $u$ and $v$, and these bounds do not match (except in some particular settings). Deriving matching lower and upper bounds should lead to a value that depends on $a$ and $b$. We will add this comment in the text.
>
> We also thank you for pointing out some typos and will correct them.

---

> > ### Comment · Reviewer_bM2e · 2024-08-08
> >
> > I thank the authors for the response. The response adequately addresses my main concerns. I encourage the authors to improve the description of the experiments to incorporate the above answers (in particular, include references to show that this is a relatively standard approach).
> >
> > My overall rating remains unchanged.

---

### Official Review · Reviewer_uKT3 · 2024-07-10

**Soundness:** 3
**Presentation:** 3
**Contribution:** 2
**Rating:** 5
**Confidence:** 4

**Summary:**

The authors analyze the metric backbone (= all the edges that are on  some shortest path) and it's relation to clustering.

They show that under weighted SBM model (with equal expected degree for all nodes), metric backbone
1) the metric backbone approximately maintains the edge probabilities of blocks
2) spectral clustering appplied to metric backbone recovers SBM.

**Strengths:**

The main contribution of the paper are the two theorems Theorem 1 and Theorem 2. Both are interesting and non-trivial results, providing connections between metric backbone and SBM models. Of these two theorems Theorem 2 is the more interesting one.

**Weaknesses:**

While theorems are interesting some assumptions seems to be very unrealistic. For example, Theorem 2 only works if tmin and tmax are the same. Is it possible to formulate it similarly as Theorem 1.

Moreover, while the edge probabilities are proportionally retained, the same (probably) doesn't hold for weights, as I assume the min cost of the cheapest path from u to v goes to 0.

Do these result also hold for other sparsification methods? such as simply deleting the edges independently with some given probability? I am guessing that Theorem 1 holds (and even stronger version) but I am not sure about Theorem 2.

Other questions/remarks:

- Assumption 1: p_n >> ...  what does >> means here exactly? that log n /  n in o(p_n)?
- Remark 1: tmin and tmax assume that there is Lambda = lambda * 1 * 1^T. Is this the assumption also in Theorem 1 and Theorem 2?
- The only difference in tmin and tmax is dmin and dmax. Why not use them directly in Theorem 1-2? Or, probably,  dmin / n and dmax / n since  those converge to a non-zero real number.
- Leveraging the memoryless ... processes. Unclear sentence, please reword.
- be defined after Equation 2.1. -> be as defined in Remark 1.
- maintains the same proportion -> maintains approximately the same proportion

**Questions:**

- Does the result hold for other sparsification methods?
- Can Theorem 2 be stated with tmin < tmax?
- is lambda in Remark 1 needed assumption?

**Limitations:**

The paper has a conclusions section but the authors do not discuss the limitations nor future work. Few sentences about the future steps would be helpful. Negative societal impact is not applicable for this paper.

---

> ### Author Rebuttal · Authors · 2024-08-05
>
> Dear Reviewer uKT3,
>
> We thank you for your time in evaluating our submission and we are grateful for your comments.
> Please find below responses to the questions raised in your reviews.
>
> Answers to the main questions:
>
> * Other sparsification methods can maintain the community structure (as shown in Figure 1) but lose other important properties such as distances and even connectivity. In weighted SBM specifically, retaining edges independently with probability $p$ preserves the community structure, resulting in a weighted SBM with the same weight distribution but lower edge density $\rho_n p$. Therefore, if $\rho_n p = \Omega( \log n / n)$, this sparsification preserves the community structure (in the sense that one could derive an analogue of Theorems 1 and 2 for this naive sparsification) but it destroys other important graph properties and structures; and notably the shortest paths will \emph{not} be preserved.
>  In contrast, our work shows that preserving the shortest paths also preserves the community structure, even though it may sacrifice other properties, such as of course the edge weight distribution.
>
> * While we assume $\tau_{\min} = \tau_{\max}$, we note that this holds true in significant settings, such as the Planted Partition Model or general SBM with the same expected degree in each community (and $\lambda_1 = \cdots = \lambda_k$). Upon closer examination of the proof of Theorem 2, we found that a weaker condition suffices, specifically $\frac{\tau_{\min}}{\tau_{\max}} = \Theta(1)$. This condition is automatically satisfied under Assumptions 1 and 2.
>  The assumption $\tau_{\min} = \tau_{\max}$ is only used twice in the proof of Theorem 2: at lines 581 and 604; all other steps already allow $\tau_{\min}$ and $\tau_{\max}$ to differ.
>  Removing strict equality in line 604 amounts to replace equation (B.6) by $\sigma_k = \frac{1}{8} \frac{ \tau_{\min} }{ \tau_{\max} } \tau \mu \log n$; if $\tau_{\min}$ and $\tau_{\max}$ are of the same order, this only changes the constant in front of the quantity $\tau \mu \log n$ and does not affect the asymptotic scaling.
>  Modifying line 581 is more involved, as we need to find the asymptotic scaling of the $k$-th largest eigenvalue of $\mathbb E W^{mb}$ without explicitly knowing the values of the matrix $\mathbb E W^{mb}$ (indeed, the upper and lower-bounds derived in Lemma 4 do not match when $\tau_{\min} \ne \tau_{\max}$). Yet, this can be handled as well; we first write $\mathbb E W_{ub}^{mb} = c_{ab}$ where $a$ and $b$ are the community of vertices $u$ and $v$. Next, we can process similarly as in the paragraph between lines 585 and 592 to derive a relationship analogous to equation (B.3).
>
>  We appreciate you raising this point, as it strengthens our result, and we plan to modify the proof and statement of Theorem 2 accordingly.
>
>
> Answers to the other remarks:
>
> * Indeed, the cost of the shortest path between $u$ and $v$ goes asymptotically to zero.
>
> * Remark 1 is here to relate $\tau_{\min}$ and $\tau_{\max}$ to the average degrees, in the particular case where $\lambda_1 = \cdots = \lambda_k$. This provides a simple intuition of what these quantities mean.  However, restricting all $\lambda_a$ to be equal is not needed in Theorems 1 and 2. When they are different, the relation between $\tau_{\min}$, $\tau_{\max}$ and the average degrees does not hold anymore, which is why we keep $\tau_{\min}$ and $\tau_{\max}$ in the main theorems.

---

### Official Review · Reviewer_Qqxh · 2024-07-12

**Soundness:** 3
**Presentation:** 3
**Contribution:** 4
**Rating:** 7
**Confidence:** 4

**Summary:**

This work focuses on the metric backbone of weighted graphs, which is the union of all-pairs shortest paths. The study analyzes the metric backbone of a class of weighted random graphs with communities and formally proves the robustness of the community structure regarding the removal of non-metric backbone edges. An empirical comparison of graph sparsification techniques also confirms the theoretical finding and shows the metric backbone is an efficient sparsifier in the presence of communities.

**Strengths:**

- Provides a formal analysis and proof to explain the robustness of the community structure in the metric backbone of weighted random graphs with communities, which fills a knowledge gap in understanding this phenomenon.
- Confirms the theoretical finding through an empirical comparison of graph sparsification techniques, providing practical evidence and validation.
- Identifies the metric backbone as an efficient sparsifier in community-present networks, which has practical applications in graph processing and analysis in various fields where community structure in networks is relevant.

**Weaknesses:**

- This sentence needs to be clearer: "This suggests that the metric backbone would dilute or destroy the community structure of the network. " What does the dilution of community structure mean? The term "preserve" may also be defined formally or mathematically.
- It claims that Algorithm 1 can recover the community structure from the metric backbone, but why the input is the original graph G?
- Although in Section 3.1, the authors claim that it is feasible to have metric backbones on networks with millions and billions of nodes, the experiments are mostly for networks with thousands of nodes. Why? Can the running time be included in the experiment?
- Comparison with other graph sparsification methods or other data sets may be added.

**Questions:**

The authors may explain the drawbacks mentioned in the weakness section.

**Limitations:**

Yes.

---

> ### Author Rebuttal · Authors · 2024-08-05
>
> Dear Reviewer Qqxh,
>
> We thank you for your time in evaluating our submission and we are grateful for your comments. Please find below responses to the questions raised in your reviews.
>
>
> * By dilution of community structure, we mean that the metric backbone could delete a larger proportion of inter-community edges than intra-community edges. We plan to make these statements more precise in the final version (see our answer to Reviewer JByP as well).
>
> * The input of Algorithm 1 is a graph; however, Theorem 2 is stated for Algorithm 1 when the input is the metric backbone of a weighted SBM. We acknowledge that this is confusing. To clarify, we plan to modify the exposition of Algorithm 1 by adding a step to compute the metric backbone $G_{mb}$ of the input graph $G$, which is then followed by performing spectral clustering on $G_{mb}$.
>
> * We discuss the running time of our implementation in Appendix C due to page limitations, but we will move this discussion to the main text in the final version. In summary, computing the metric backbone is faster than spectral sparsification (where we used an external library), and running spectral clustering took significantly longer.
>
> * We compared the metric backbone with the threshold graph (a naive but still commonly used technique) and spectral sparsification (one of the most popular and efficient graph sparsification techniques). We also provide additional experiments in Appendix C. This comparison demonstrates that the metric backbone is a competitive sparsifier. The goal of this paper is to highlight that the metric backbone preserves community structure, which is why we limited the baselines for comparison. A comprehensive review and comparison of all graph sparsification techniques could be the topic of an interesting review paper but would go beyond the scope of the current paper.

---

### Official Review · Reviewer_JByP · 2024-07-12

**Soundness:** 3
**Presentation:** 3
**Contribution:** 3
**Rating:** 7
**Confidence:** 4

**Summary:**

The authors investigate the shortest-path backbone that is the union of all-pair shortest paths in a weighted graph, and show that it preserves the community structure in weighted Stochastic Block Model (wSBM) with high probability. A key finding in their proof is that the metric backbone maintains the same proportion of intra- and inter- cluster edges as in the original wSBM graph. Then the classic Spectral Clustering algorithm is able to reconstruct the embedded communities with high probability. Their experimental results confirm that the shortest-path backbone can preserve community structures in real datasets, and is also an efficient sparsifier in the presence of communities.

**Strengths:**

1. The technical contribution is solid.
2. The paper is well written and the presentation of proof ideas is clear.

**Weaknesses:**

The theoretical results seems to heavily depend on the property of wSBM, and are hardly generalized to much wider well-clustered graphs. It is possibly more interesting to figure out in what conditions the metric backbone is able to preserve communities in a given well-clustered graph. This is a much closer scenario to the observations in the experiments.

**Questions:**

1. I am not sure whether it is proper to say that it is intuitive that metric sparsification should dilute the community structure (in Line 56), since for the simplest case of unweighted graphs, the metric backbone clearly maintains all edges and the community structure will be preserved certainly.

2. It seems that the Spectral Clustering algorithm (Algorithm 1) deals directly with the edge weights of wSBM that is introduced by the exponential distribution for a distance metric. But the Spectral clustering process works only for similarity-based graphs. Why don’t the authors need to change the distance metric into a proximity one before Spectral Clustering?

**Limitations:**

The authors haven’t discussed the limitations in the main paper, but in the checklist instead.

---

> ### Author Rebuttal · Authors · 2024-08-05
>
> Dear Reviewer JByP,
>
> We thank you for your time in evaluating our submission and we are grateful for your comments. Please find below responses to the questions raised in your reviews.
>
>
> * Intuitively, keeping only the shortest paths and deleting everything else could destroy all structures unrelated to the shortest paths (such as the communities). This is indeed not the case on unweighted graphs, as the metric backbone does not remove any edge, hence the whole structure is preserved. However, community structure and shortest paths appear to act as two antagonists in graph sparsification, because removing edges that belong to a large number of shortest paths destroys these shortest paths (and graph connectivity) but can reveal communities as disconnected components of the sparsified graph (see ref. [17]).
>  Moreover, when the metric backbone keeps only a small fraction of the edges of the original weighted graph, one could expect that the fractions of inter- and intra-community edges might be affected differently by the deletion of non-shortest path edges. Our work shows that this is not the case.
>
> * We acknowledge that spectral clustering is an umbrella term that encompasses various techniques involving the eigenstructure of a matrix characterizing a graph topology. Spectral clustering on the normalized Laplacian of the graph (which is the version implemented in popular libraries such as scikit-learn) requires weights to be similarities between node pairs and not distances. Because we are directly working with the graph adjacency matrix, we can handle weights being either similarities or distances. Nonetheless, we also acknowledge a typo in line 2 of Algorithm 1: the eigenvalues should be ordered in decreasing *absolute* value (note that this is already the case in the proof, see for example line 582).
>
>  As a naive justification of why ordering in absolute value handles all cases, consider an unweighted SBM with two communities of the same size, and the probability of intra- (resp., inter-) community edge $p$ (resp., $q$). The eigenvalues of $\mathbb E A$ are $n(p+q)/2$ (multiplicity 1), $n(p-q)/2$ (multiplicity 1) and $0$ (multiplicity $n-2$). When $p\ne q$, we indeed have $|n(p+q)/2| > |n(p-q)/2| > 0$. But the relationship $n(p+q)/2 > n(p-q)/2 > 0$ does not hold if $p<q$.
>
>
> [17] Michelle Girvan and Mark EJ Newman. Community structure in social and biological networks. Proceedings of the national academy of sciences, 99(12):7821–7826, 2002.

---

> > ### Comment · Reviewer_JByP · 2024-08-13
> >
> > Thank the authors for their response. I keep my score.

---

### Decision · Program_Chairs · 2024-09-25

**Decision:**

Accept (poster)

**Comment:**

There is general agreement among the reviewers that this submission is a technically solid contribution to network theory with high impact and good evaluations.  In particular, the authors prove a robustness result about the metric backbone of weighted random graphs that had been observed experimentally but so far without a proof.  They further provide additional experimental validation of their result.